# Using Interleaved Ensemble Unlearning to Keep Backdoors at Bay for Finetuning Vision Transformers

## Abstract

Vision Transformers (ViTs) have become popular in computer vision tasks. Backdoor attacks, which trigger undesirable behaviours in models during inference, threaten ViTs' performance, particularly in security-sensitive tasks. Although backdoor defences have been developed for Convolutional Neural Networks (CNNs), they are less effective for ViTs, and defences tailored to ViTs are scarce. To address this, we present Interleaved Ensemble Unlearning (IEU), a method for finetuning clean ViTs on backdoored datasets. In stage 1, a shallow ViT is finetuned to have high confidence on backdoored data and low confidence on clean data. In stage 2, the shallow ViT acts as a "gate" to block potentially poisoned data from the defended ViT. This data is added to an unlearn set and asynchronously unlearnt via gradient ascent. We demonstrate IEU's effectiveness on three datasets against 11 state-of-the-art backdoor attacks and show its versatility by applying it to different model architectures.

## 1 Introduction

Vision Transformers (ViTs, Dosovitskiy et al. (2021)) have emerged as a powerful alternative to Convolutional Neural Networks (CNNs) for a wide range of computer vision tasks. ViTs have achieved state-of-the-art performance in various downstream tasks such as image classification, object detection, and semantic segmentation (Thisanke et al., 2023; Shehzadi et al., 2023). However, the widespread deployment of ViTs have also raised concerns about their vulnerability to adversarial threats, particularly backdoor attacks, which typically modify images and/or labels in the training dataset to trigger attacker-controlled undesirable behaviour during inference (Gu et al., 2019; Subramanya et al., 2024; Yuan et al., 2023). Backdoor attacks such as the BadNets attack in Gu et al. (2019) can compromise model behaviour by embedding malicious triggers during training, leading to security risks in real-world applications. As ViTs become increasingly popular in security-sensitive domains such as autonomous driving and face recognition (Lai-Dang, 2024; Tran et al., 2022), it is important to understand these vulnerabilities and develop robust backdoor defences for ViTs.

ViTs are often pretrained using self-supervised learning (SSL) on large datasets and then finetuned to be deployed on specific tasks. Backdoor defences have been proposed to defend foundation models pretrained on large datasets and can either prevent backdoor injection during the SSL process or encourage removal of backdoors after pretraining (Tejankar et al., 2023; Bie et al., 2024); these thwart backdoor attacks that occur during pretraining, such as a practical real-world attack on web-scraped datasets in Carlini et al. (2024) and an SSL-specific imperceptible attack Zhang et al. (2024). The finetuning process for adapting ViTs to downstream tasks using supervised learning is equally vulnerable to backdoor attacks. The rationale behind developing ViT-specific defences for finetuning are two-fold: Mo et al. (2024) shows that there are few defences specifically designed for ViTs for image classification in existing literature (Doan et al. (2023) and Subramanya et al. (2024) being notable examples of such defences); in addition, although existing defences designed for CNNs can defend ViTs after modifying the defence implementations, they still lead to high ASR and/or low CA when defending different flavours of ViTs (Tables 4 & 5 in Mo et al. (2024)).

To fill the gap, this work propose a novel backdoor defence that uses an ensemble of two ViTs to perform **interleaved unlearning** on potentially poisoned data, demonstrating *superior performance*

Figure 1: Overview of our defence, **IEU**. The red poisoned module and blue robust module are represented by $f_p$ and $f_r$, respectively. Shaded boxes are conditions; underlined text represent actions. The lock icon indicates a frozen network. The blue network is shielded from poisoned images by the red network and the blue network unlearns potentially poisoned data. Unaugmented images are used for $f_p$ during both stages. Please refer to Section 3 for a summary of notations.

*on ViTs* compared to previous SOTA methods. Informed by the designs in Liu et al. (2024) and Li et al. (2021b), we use a shallow ViT denoted by the "poisoned module" to defend the main ViT, which we call the "robust module". Our design **IEU** has two stages as shown in Figure 1. In **stage 1**, the poisoned module ($f_p$), a shallow ViT, is tuned on the attacker-controlled finetuning data. Intuitively, shortcut learning (Geirhos et al., 2020) leads $f_p$ to learn shortcuts in the dataset, which are most prevalent in poisoned images. In addition, the simplicity of $f_p$ discourages it from learning clean data that have fewer shortcuts. Therefore, the poisoned module is confident (where confidence is the maximum class probability $\max[\sigma(\hat{\mathbf{y}}_p)]$ predicted by $f_p$) when classifying poisoned data and not confident otherwise. In **stage 2**, images pass through the tuned $f_p$, which either queues data onto the unlearn set or allows the defended main ViT (the robust module) to learn data normally based on the confidence threshold $c_{\text{thresh}}$. Whenever the unlearn set accumulates enough potentially poisoned data, a batch is unlearnt by the robust module using a dynamic unlearning rate. Instead of using a pre-determined unlearn set, our defence accumulates the unlearn set during stage 2. The benefits are two-fold: compared to using ABL's (Li et al., 2021b) method which isolates poisoned samples using the defended model, finetune-time unlearn set accumulation using $f_p$ ensures that the robust module learns as little poisoned data as possible; in addition, online accumulation of $\mathcal{D}^{\text{ul}}$ is adaptive in the sense that the frequency of unlearning is high when more potentially poisoned images are encountered, quickly erasing the impact of a large number of poisoned data. In addition, we argue that core concepts developed in our method, namely applying interleaved unlearning, can defend other model architectures in image classification. Here are our main contributions:

- We propose the universally applicable and novel **interleaved unlearning framework** as a backdoor defence. The defence, incorporated into IEU, alternates between learning benign data and unlearn backdoored data. We show that our IEU is successful without requiring high-precision isolation of poisoned data and performs especially well on ViTs.
- We empirically demonstrate that our design out-performs existing state-of-the-art defences on challenging datasets using **11 backdoor attacks** by comparing to SOTA methods such as ABL and I-BAU (Li et al., 2021b; Zeng et al., 2021b); Attack Success Rate (ASR) improved by 33.83% and 31.46% on average for TinyImageNet and CIFAR10, respectively, while maintaining high Clean Accuracy (CA).
- We **demonstrate IEU's universality** by successfully defending ViT variants and CNN architectures (Table 8). Furthermore, we show that IEU successfully repels an **adaptive attack** (Table 18).
- We explore potential points of failure of unlearning-based defence mechanisms to defend against *weak* attacks where "weakness" corresponds to lower ASR. We propose potential solutions to address these failures. In our opinion, weak attacks are as insidious as powerful attacks.

## 2 RELATED WORK

**Backdoor Attacks**. Attackers aim (a) to induce a specific classification when the input is perturbed by an attacker-specified transformation and (b) to maintain normal performance when images without a backdoor trigger are classified (Gu et al., 2019). Often, the attacker achieves the two goals by injecting poisoned images into the training set. Backdoor attacks for both SSL (Zhang et al., 2024; Sun et al., 2024; Saha et al., 2022; Li et al., 2023a; Jia et al., 2022) and supervised learning (Chen et al., 2017; Liu et al., 2018b; Nguyen & Tran, 2021; Li et al., 2021a) have been proposed. There

are three categories of backdoor attacks for supervised learning, which is the learning phase that this work defends: dirty-label attacks which includes visible and invisible attacks (Tan & Shokri, 2020; Lin et al., 2020; Doan et al., 2021), clean-label attacks which do not modify the label of backdoored images (Gao et al., 2023b; Zeng et al., 2023; Turner et al., 2019), and clean-image attacks which only modify data labels (Rong et al., 2024; Chen et al., 2023). Authors have also developed *ViT-specific* backdoor attacks. For example, Zheng et al. (2023) inserts a Trojan into a ViT checkpoint, while Lv et al. (2021) modifies the finetuning procedure by using an attacker-specified loss function.

**Backdoor Defences**. Defenders aim to ensure that backdoor images do not trigger attacker-specified model behaviour whilst maintaining high CA. A popular class of defences is model reconstruction where defenders cleanse poisoned models of backdoors (Liu et al., 2017; 2018a). Works in this category aim to remove backdoor neurons and include Neural Attention Distillation (NAD, Li et al. (2021c)), Adversarial Neuron Pruning (ANP, Wu & Wang (2021)) Adversarial Weight Masking (AWM, Chai & Chen (2022)), Shapley-estimation based few-shot defence (Guan et al., 2022), and Reconstructive Neuron Pruning (RNP, Li et al. (2023b)). Another such cleansing defence, I-BAU (Zeng et al., 2021b) connects the two optimisation problems in the minimax formulation of backdoor removal using an implicit hypergradient. Certified backdoor defences have also been developed (Weber et al., 2023). Another broad class of defences involves reconstructing the trigger in order to unlearn backdoor images. Notable examples include Neural Cleanse (Wang et al., 2019), DeepInspect (Chen et al., 2019) which checks for signs of backdooring without a reserved clean set, and BTI-DBF (Xu et al., 2024) which decouples benign features for backdoor trigger inversion. Tuning clean models on backdoored datasets (Borgnia et al., 2021; Wang et al., 2022a;b; Zhang et al., 2023) is also popular and is most related to our IEU. Additionally, methods such as Anti-Backdoor Learning (ABL, Li et al. (2021b)) and ASD (Gao et al., 2023a) focus on isolating poisoned data.

**ViT-Specific Backdoor Defences**. Few backdoor defences are specifically designed for defending ViTs during tuning (Mo et al., 2024) and existing defences that have CNNs in mind perform worse on ViTs. Two notable defences are Doan et al. (2023) where backdoor images are identified using patch-processing, and Subramanya et al. (2024) which is a test-time defence that uses GradRollout (Gildenblat (2020), an interpretation method for ViTs) to block high-attention patches in images.

**Machine Unlearning** (Xu et al., 2023a) focuses on removing data from models due to privacy reasons; additionally, unlearning is also useful for removing unwanted associations between certain undesirable features and labels, making it useful for backdoor defence as shown in Li et al. (2021b).

## 3 METHOD

In this section, we first define our threat model and describe IEU in detail. We conclude this section by briefly exploring the drawbacks of using a fixed-size unlearn set $\mathcal{D}^{\text{ul}}$.

**Threat model**. We focus on finetuning ViTs for image classification tasks and assume that the pretrained model checkpoint initially given to the defender is not benign. We follow the threat model of Li et al. (2021b). We assume that the finetuning procedure is controlled by the defender, which means attacks that modify finetuning loss or the model's gradient (Lv et al., 2021; Bagdasaryan & Shmatikov, 2021) are out of scope. On the other hand, finetuning data is gathered from untrusted sources and may contain backdoor data. The attacker knows the model architecture and the pretrained checkpoint's parameter values, and may poison the finetuning dataset by modifying images and/or labels. The defender aims to tune a benign checkpoint for downstream tasks using $\mathcal{D}^{\text{tune}}$ and does not know the distribution/proportion of backdoor data in the attacker-supplied $\mathcal{D}^{\text{tune}}$.

**Notations**. The finetuning set $\mathcal{D}^{\text{tune}}$ *may* contain an unknown proportion of backdoor samples; this proportion (i.e., poisoning rate) is denoted by $\alpha$. The defender unlearns data in the unlearn set $\mathcal{D}^{\text{ul}}$, whose size as a fraction of $\mathcal{D}^{\text{tune}}$ is defined as $\hat{\alpha} = |\mathcal{D}^{\text{ul}}| \div |\mathcal{D}^{\text{tune}}|$. The two sub-networks in the ensemble are the poisoned module and robust module, denoted by $f_p$ and $f_r$, respectively. Data points $(\mathbf{x}, \mathbf{y}) \in \mathcal{D}^{\text{tune}}$, which consist of unaugmented ($\mathbf{x}_{\text{noAug}}$) and augmented ($\mathbf{x}_{\text{yesAug}}$) views of the original image (as in "data augmentation"), and potentially poisoned data points $(\mathbf{x}^{\hat{p}}, \mathbf{y}^{\hat{p}}) \in \mathcal{D}^{\text{ul}}$ are used to finetune and defend $f_r$, respectively. For simplicity, we use $\mathbf{x}$ to denote images when data augmentation is not relevant. The logits produced by the two modules are referred to as $\hat{\mathbf{y}}_p = f_p(\mathbf{x}; \boldsymbol{\theta}_p)$ and $\hat{\mathbf{y}}_r = f_r(\mathbf{x}; \boldsymbol{\theta}_r)$, where $\boldsymbol{\theta}_p, \boldsymbol{\theta}_r$ are the potentially tunable parameters of $f_p$ and $f_r$, respectively. The two logits vectors $\hat{\mathbf{y}}_p$ and $\hat{\mathbf{y}}_r$ combine to form the logits vector $\hat{\mathbf{y}}$ based on $m_{\boldsymbol{\theta}_p}$

(Equation 1). We use $\sigma(\cdot)$ and $\ell(\cdot, \cdot)$ to represent the softmax function and the cross-entropy loss, respectively. The confidence threshold $0 < c_{\text{thresh}} < 1$ determines whether an image is asynchronously unlearned or immediately learned. The number of classes in $\mathcal{D}^{\text{tune}}$ is denoted by $N_c$ for CIFAR10, GTSRB, and TinyImageNet, respectively. The learning rates used to finetune $f_r$ and unlearn $\mathbf{x}^{\hat{p}}$ are $lr^{\text{tune}}$ and $lr^{\text{ul}}$, respectively.

### 3.1 INTERLEAVED ENSEMBLE UNLEARNING (**IEU**)

**Overview.** Figure 1 summarises our method, which has two stages. During stage 1, the poisoned module $f_p(\cdot; \boldsymbol{\theta}_p)$ is pre-finetuned using finetuning data $\mathcal{D}^{\text{tune}}$. During stage 2, $f_p$ is used to determine whether incoming data is learned by the robust module $f_r(\cdot; \boldsymbol{\theta}_r)$ or added to $\mathcal{D}^{\text{ul}}$ for asynchronous unlearning based on the unlearn rate $lr^{\text{ul}}$ in Equation 3.

**Stage 1: Isolating backdoored data** by pre-finetuning the poisoned module $f_p$. This step applies vanilla tuning (for hyperparameters see Table 11) using $\mathcal{D}^{\text{tune}}$ on $f_p(\cdot; \boldsymbol{\theta}_p)$, where $\boldsymbol{\theta}_p$ is initialised as the first few layers of a pretrained checkpoint. Stage 1 solves the following optimisation problem: $\min_{\boldsymbol{\theta}_p} \mathbb{E}_{\mathbf{x}_{\text{noAug}} \sim \mathcal{D}^{\text{tune}}}[\ell(f_p(\mathbf{x}_{\text{noAug}}; \boldsymbol{\theta}_p), \mathbf{y})]$, where $\ell(\cdot, \cdot)$ is the cross entropy loss, $\mathbf{y}$ is the one-hot ground truth vector, and $f_p$ is the poisoned module. As explained in Section 1, the intuition of overfitting $f_p$ on poisoned data is based on shortcut learning (Geirhos et al., 2020). These shortcuts are found in backdoored images, where the attacker-specified trigger acts as an easily identifiable artifact that causes $f_p$ to easily learn the connection between the trigger and attacker-specified label. Therefore, the tuned $f_p$ is likely to be confident when predicting images with the trigger. The poisoned module $f_p$ is designed to be complex enough to learn shortcuts and shallow enough to avoid learning much from benign data. The goal is for $\max(\sigma(f_p(\mathbf{x}_{\text{noAug}}; \boldsymbol{\theta}_p)))$ to be small for clean images and large for backdoor images after stage 1. Note that the poisoned module is replaceable by other methods (Doan et al., 2023; Li et al., 2021b; Gao et al., 2023a) that isolate backdoored data and $f_p$ is not absolutely necessary for interleaved unlearning.

**Stage 2: Apply Interleaved Unlearning** on the robust module $f_r$. This stage optimises the objective in Equation 2, minimising the loss on clean data and maximising the loss on poinsoned data; $\boldsymbol{\theta}_r$ is initialised as a pretrained checkpoint. During this stage, $\boldsymbol{\theta}_p$ is frozen and only $\boldsymbol{\theta}_r$ is tuned. The logits $\hat{\mathbf{y}}_p = f_p(\mathbf{x}_{\text{noAug}}; \boldsymbol{\theta}_p)$ for the unaugmented view of each image is produced by $f_p$ in order to compute maximum class probability $\max(\sigma(\hat{\mathbf{y}}_p))$, which is then compared to $c_{\text{thresh}}$ to determine whether the data point should be learned by $f_r$ or added onto $\mathcal{D}^{\text{ul}}$. If $\max(\sigma(\hat{\mathbf{y}}_p))$ is above $c_{\text{thresh}}$, the data point is added to $\mathcal{D}^{\text{ul}}$. Otherwise, $f_r$ learns the augmented views as in regular finetuning. To prevent poisoned data from being learned, we apply *logit masking* on the output logits of $f_r$ and $f_p$ (Equation 1)

$$m_{\boldsymbol{\theta}_p} = \mathbf{1}_{x < c_{\text{thresh}}}(\max(\sigma(f_p(\mathbf{x}_{\text{noAug}}; \boldsymbol{\theta}_p)))), \text{ where } \hat{\mathbf{y}} = \hat{\mathbf{y}}_p(1 - m_{\boldsymbol{\theta}_p}) + \hat{\mathbf{y}}_r m_{\boldsymbol{\theta}_p} \tag{1}$$

where $m_{\boldsymbol{\theta}_p}$ is the binary logit mask, $\mathbf{1}_{x < c_{\text{thresh}}}(x)$ is the indicator function, $\mathbf{y}$ is the ground truth, $\hat{\mathbf{y}}$ is the logits vector, and $\hat{\mathbf{y}}_p = f_p(\mathbf{x}_{\text{noAug}}; \boldsymbol{\theta}_p)$, $\hat{\mathbf{y}}_r = f_r(\mathbf{x}_{\text{yesAug}}; \boldsymbol{\theta}_r)$ are the logits produced by $f_p$ and $f_r$, respectively. When $\mathbf{x}_{\text{noAug}}$ is detected by $f_p$ as a potentially poisoned image, the logits for optimising the "Learning" objective in Equation 2 come from $f_p$; otherwise, $\hat{\mathbf{y}} = \hat{\mathbf{y}}_r$. In other words, optimising the "Learning" objective (Equation 2) requires both $f_p$ and $f_r$ to contribute to the logits.

$$\min_{\boldsymbol{\theta}_r} \underbrace{\mathbb{E}_{\mathbf{x} \sim \mathcal{D}^{\text{tune}}}[\ell(\hat{\mathbf{y}}, \mathbf{y})]}_{\text{Learning}} - \underbrace{\mathbb{E}_{\mathbf{x}^{\hat{p}} \sim \mathcal{D}^{\text{ul}}}[\ell(f_r(\mathbf{x}^{\hat{p}}; \boldsymbol{\theta}_r), \mathbf{y}^{\hat{p}})]}_{\text{Unlearning}} \tag{2}$$

The unlearn set $\mathcal{D}^{\text{ul}}$ accumulates data until it has enough data for one batch containing potentially poisoned images $(\mathbf{x}^{\hat{p}}, \mathbf{y}^{\hat{p}})$, which is then unlearnt by $f_r$ during finetuning. Unlike the continuously decaying learning rate $lr^{\text{tune}}$ used for normal finetuning, the unlearning rate $lr^{\text{ul}}$ doesn't depend on just the decay schedule. Given $lr^{\text{tune}}$ which follows the cosine annealing decay schedule, the $(k-1)^{\text{th}}$ batch with potentially poisoned images $(\mathbf{x}^{\hat{p}}_{k-1}, \mathbf{y}^{\hat{p}}_{k-1})$, the number of classes in the dataset $N_c$, and the robust module $f_r(\cdot; \boldsymbol{\theta}_r)$, the dynamic unlearning rate for the current batch $(\mathbf{x}^{\hat{p}}_k, \mathbf{y}^{\hat{p}}_k)$ is defined in Equation 3 and can be viewed as a function of cross entropy loss of the previous batch of potentially poisoned images $\ell(f_r(\mathbf{x}^{\hat{p}}_{k-1}; \boldsymbol{\theta}_r), \mathbf{y}^{\hat{p}}_{k-1})$.

$$lr^{\text{ul}}_k = lr^{\text{tune}} \cdot \left( \mathbf{1}_{k>0}(k) \cdot \max\left[ 6 - \exp\left[ -\left( \ln(N_c) - \ell(f_r(\mathbf{x}^{\hat{p}}_{k-1}; \boldsymbol{\theta}_r), \mathbf{y}^{\hat{p}}_{k-1}) \div \sqrt{2} \right) \right], 0.2 \right] + \mathbf{1}_{k=0}(k) \right) \tag{3}$$

The two indicator functions ensure $lr^{\text{ul}}_k = lr^{\text{tune}}$ when $k = 0$, which occurs at an epoch's beginning. The term that scales $lr^{\text{tune}}$ in Equation 3 is an exponentially decreasing function with respect to increasing loss $\ell(f_r(\mathbf{x}^{\hat{p}}_{k-1}; \boldsymbol{\theta}_r), \mathbf{y}^{\hat{p}}_{k-1})$, causing $lr^{\text{ul}}_k$ to be large when the previous batch $\mathbf{x}^{\hat{p}}_{k-1}$ produces

low cross entropy loss on $f_r$. This keeps the backdoor from being learned by $f_r$. Although it is shown in Table 9 that using $lr_k^{\text{ul}} = c \cdot lr^{\text{tune}}$ for some $c \in \mathbb{R}^+$ performs better than using Equation 3, one benefit of defining $lr_k^{\text{ul}}$ using a fixed function is that $lr_k^{\text{ul}}$ is no longer a hyperparameter that needs to be tuned. In addition to the "Learning" objective, interleaved unlearning optimises the "Unlearning" objective in Equation 2, which is implemented using gradient ascent performed on $f_r$ given $(\mathbf{x}^{\hat{p}}, \mathbf{y}^{\hat{p}}) \in \mathcal{D}^{\text{ul}}$. See Algorithm 1 in Appendix B for a precise description of stage 2.

## 3.2 WHY NOT A FIXED-SIZED UNLEARN SET?

In this subsection, we argue that, for IEU, isolating a variable fraction of the training set as the poisoned set leads to better performance. Authors in ABL (Li et al., 2021b) isolate a fixed fraction (called "$r_{\text{isol}}$") of the tuning set where $0 \le r_{\text{isol}} \le 1$ (they used $r_{\text{isol}} = 0.01$). In their method, images whose cross entropy loss rank amongst the lowest $r_{\text{isol}}$ fraction of $\mathcal{D}^{\text{tune}}$ is collected to form $\mathcal{D}^{\text{ul}}$ of size $\hat{\alpha} = r_{\text{isol}}$. ABL uses techniques such as Local Gradient Ascent (LGA) or loss Flooding *on the defended model* to encourage poisoned images to have low loss. Compared to using $r_{\text{isol}}$ (ABL), there are two main advantages for using $c_{\text{thresh}}$ (our method) to produce the unlearn set in our method: (a) the effectiveness (FPR or FNR) of our isolation method is not significantly affected by the value of the poisoning rate $\alpha$, which is unknown to the defender (Table 1), and (b) the unlearn set size varies as $\alpha$ varies, which increases defence success using IEU since a high proportion of poisoned data should be added to $\mathcal{D}^{\text{ul}}$ for low ASR and high CA ($\hat{\alpha}_i \div \alpha \in \{0.9, 1.0\}$ in Table 2).

Table 1: Performance of the two methods when evaluated on detecting poisoned finetuning data (CIFAR10). Five *poisoned module* ($f_p$) instances are pre-finetuned for 10 epochs at $2 \cdot 10^{-4}$ learning rate with different poisoning rate $\alpha$; $c_{\text{thresh}}$ and $r_{\text{isol}}$ are fixed at 0.95 and 0.1, respectively. Each cell shows the FPR/FNR values as percentages ("positive" means "poisoned").

| Attack | Selection Method | $\alpha = 0.02$ | 0.05 | 0.10 | 0.15 | 0.20 |
|---|---|---|---|---|---|---|
| BadNets-white | $r_{\text{isol}}$ | 8.68/25.30 | 6.42/22.04 | 0.80/7.22 | 0.03/33.49 | 0.00/50.00 |
| | $c_{\text{thresh}}$ | 9.16/24.90 | 5.67/22.84 | 5.62/5.30 | 6.09/10.25 | 4.71/5.17 |
| ISSBA | $r_{\text{isol}}$ | 8.59/21.00 | 5.60/6.44 | 0.62/5.54 | 0.00/33.33 | 0.00/50.00 |
| | $c_{\text{thresh}}$ | 6.27/25.40 | 6.79/5.76 | 5.55/2.58 | 6.51/1.48 | 5.68/0.48 |

Table 2: Performance of models that are defended during stage 2 using IEU with hand-crafted unlearn sets $\mathcal{D}_i^{\text{ul}}$ of varying sizes ($\hat{\alpha}_i$) as a fraction of the *original* finetune set $\mathcal{D}^{\text{tune}}$. The poisoning rate is fixed at $\alpha = 0.1$ and the sizes of $\mathcal{D}_i^{\text{ul}}$ as a fraction of $\mathcal{D}^{\text{tune}}$ are $\hat{\alpha}_i \in (0.01, 0.02, 0.05, 0.1, 0.2)$. A hand-crafted unlearn set $\mathcal{D}_i^{\text{ul}}$ consists entirely of poisoned data if $\hat{\alpha}_i \div \alpha \le 1$ and includes all poisoned data if $\hat{\alpha}_i \div \alpha \ge 1$. All values here are expressed in percentages.

| Size ratio ($\hat{\alpha}_i \div \alpha$): | | 0.1 | | 0.2 | | 0.5 | | 0.9 | | 1.0 | | 2.0 | |
|---|---|---|---|---|---|---|---|---|---|---|---|---|---|---|
| Dataset | Attack | ASR | CA | ASR | CA | ASR | CA | ASR | CA | ASR | CA | ASR | CA |
| | BadNets-white | 10.09 | 98.18 | 9.89 | 97.58 | 6.41 | 96.30 | 0.88 | 97.94 | 0.88 | 98.28 | 0.67 | 97.83 |
| CIFAR10 | ISSBA | 100.00 | 98.31 | 100.00 | 98.18 | 0.00 | 96.94 | 0.00 | 97.99 | 0.00 | 98.23 | 0.00 | 98.16 |
| | Smooth | 95.32 | 98.24 | 82.28 | 97.93 | 0.06 | 97.00 | 0.77 | 97.98 | 0.94 | 98.24 | 0.11 | 97.90 |
| | BadNets-white | 0.32 | 61.95 | 0.01 | 57.89 | 0.00 | 54.03 | 0.00 | 63.43 | 0.00 | 65.95 | 0.00 | 37.18 |
| TinyImageNet | ISSBA | 57.84 | 64.94 | 0.15 | 58.70 | 0.00 | 22.50 | 0.00 | 38.63 | 0.16 | 46.97 | 0.05 | 16.37 |
| | Smooth | 93.51 | 68.37 | 77.33 | 64.84 | 0.00 | 55.93 | 0.00 | 61.95 | 0.01 | 66.28 | 0.00 | 38.86 |

Table 1 shows that the FPR/FNR are similar for both $r_{\text{isol}}$ and $c_{\text{thresh}}$ at low poisoning rate. However, as $\alpha$ increases, using $r_{\text{isol}}$ causes more poisoned data to be left out of $\mathcal{D}^{\text{ul}}$. For example, at $\alpha = 0.2$ and $r_{\text{isol}} = 0.1$ (meaning $\hat{\alpha}_i \div \alpha = 0.1 \div 0.2 = 0.5$), the FNR is 50%. This results in instability during defence and worse performance as shown in Table 2 (column 0.5) since a large fraction of poisoned data is not in $\mathcal{D}^{\text{ul}}$. As less poisoned data is included in $\mathcal{D}^{\text{ul}}$, our defence becomes less effective with ASR increasing and CA decreasing (Table 2). Since using a fixed $r_{\text{isol}}$ leaves many poisoned images outside of $\mathcal{D}^{\text{ul}}$ when $\alpha > r_{\text{isol}}$, we use $c_{\text{thresh}}$ to select a variable-sized $\mathcal{D}^{\text{ul}}$.

We show in Table 15 (Appendix C) that our shallow $f_p$ is not compatible with LGA/Flooding when tuning with CIFAR10/TinyImageNet; however, applying LGA/Flooding during stage 1 is helpful when using IEU with GTSRB.

## 4 EXPERIMENTS

See Appendix A for more details regarding baselines, attacks, datasets and defence parameters.

**Baselines**. We use three baseline methods for comparison with our defence. Specifically, we compare against I-BAU (Zeng et al., 2021b), ABL (Li et al., 2021b), and AttnBlock (Subramanya et al., 2024) which is a ViT-specific defence. We report results for AttnBlock in Appendix A.2 due to high ASR. I-BAU and ABL are state-of-the-art general defences not specifically designed for ViTs; authors in (Wang et al., 2022a) suggest that I-BAU is the most competitive baseline compared to others. We attempt and fail to reproduce the RNP defence (Li et al., 2023b) for ViTs despite following recommendations in Mo et al. (2024) to mask features of linear layers instead of those of norm layers.

**Attacks**. We evaluate the performance of our design on 11 backdoor attacks. Specifically, we consider 9 out of 10 attacks in Wang et al. (2022a): BadNets-white (white lower-right corner), BadNets-pattern (grid pattern in lower-right corner) (Gu et al., 2019), Blended (Chen et al., 2017), l0-inv, l2-inv (Li et al., 2021a), Smooth (Zeng et al., 2021a), Trojan-SQ, Trojan-WM (Liu et al., 2018b), and a clean label attack, SIG (Barni et al., 2019). In addition, we consider the sample-specific invisible attack ISSBA (Li et al., 2021d) and an image transformation-based attack BATT (Xu et al., 2023b). Please

Table 3: Performance of IEU compared to no defence, ABL (Li et al., 2021b), I-BAU (Zeng et al., 2021b) given as percentages. Averages of each column are given in the last row for that dataset and best/second-best values are bolded/underlined. See Appendix A.2 for AttnBlock results.

| Dataset | Attack | No Defence | | I-BAU | | ABL | | IEU (ours) | |
|---|---|---|---|---|---|---|---|---|---|
| | | ASR | CA | ASR | CA | ASR | CA | ASR | CA |
| CIFAR10 | BadNets-white | 97.51 | 98.36 | 10.12 | 95.86 | 9.7 | 98.11 | **0.96** | **98.19** |
| | BadNets-pattern | 100.0 | 98.23 | 91.84 | 92.16 | 100.0 | 97.4 | **0.0** | **98.22** |
| | ISSBA | 100.0 | 98.13 | 9.46 | 92.96 | 100.0 | 37.62 | **0.33** | **98.35** |
| | BATT | 100.0 | 98.28 | 21.68 | 95.86 | 90.05 | 97.94 | **0.02** | **98.23** |
| | Blended | 100.0 | 98.39 | 21.0 | 93.92 | 25.04 | 97.82 | **0.0** | **98.27** |
| | Trojan-WM | 99.99 | 98.32 | 79.94 | 94.16 | 99.91 | 97.97 | **0.0** | **98.15** |
| | Trojan-SQ | 99.7 | 98.31 | 68.26 | 94.62 | 99.62 | **98.27** | **0.04** | 98.22 |
| | Smooth | 99.76 | 98.24 | 16.26 | 94.0 | 9.31 | 97.12 | **0.09** | **97.77** |
| | l0-inv | 100.0 | 98.34 | 10.28 | 93.24 | 0.04 | 97.12 | **0.0** | **98.19** |
| | l2-inv | 99.98 | 98.41 | 9.98 | 93.4 | 8.62 | 98.1 | **0.44** | **98.24** |
| | SIG | 98.49 | 88.75 | 9.16 | **94.38** | 97.94 | 88.44 | **0.0** | 87.67 |
| | **Average** | 99.58 | 97.43 | 31.63 | **94.05** | 58.2 | 91.45 | **0.17** | 97.23 |
| GTSRB | BadNets-white | 95.7 | 95.63 | 5.48 | **99.1** | 4.09 | 92.83 | 2.22 | 83.26 |
| | BadNets-pattern | 100.0 | 96.44 | 5.46 | **98.17** | 4.09 | 93.76 | **0.0** | 95.11 |
| | ISSBA | 99.99 | 95.91 | 5.48 | **99.48** | 3.63 | 93.15 | 2.81 | 86.48 |
| | BATT | 100.0 | 96.18 | 6.08 | **99.58** | **2.29** | 92.79 | 7.4 | 88.38 |
| | Blended | 100.0 | 96.95 | 11.24 | **97.75** | **0.0** | 92.95 | 6.83 | 89.25 |
| | Trojan-WM | 100.0 | 92.75 | 5.26 | **98.11** | **0.0** | 75.16 | 8.87 | 91.81 |
| | Trojan-SQ | 99.85 | 94.91 | 5.48 | **99.61** | 99.6 | 81.99 | **2.85** | 89.02 |
| | Smooth | 99.79 | 96.29 | 27.5 | **99.74** | 3.46 | 92.36 | 15.37 | 88.45 |
| | l0-inv | 100.0 | 96.76 | 5.44 | **99.7** | 100.0 | 94.96 | **0.0** | 88.81 |
| | l2-inv | 100.0 | 93.93 | **9.06** | **99.71** | 77.79 | 96.03 | 20.26 | 83.67 |
| | SIG | 99.52 | 91.41 | 2.92 | 38.32 | 95.53 | **83.49** | **0.0** | 77.39 |
| | **Average** | 99.53 | 95.2 | 8.13 | **93.57** | 35.5 | 89.95 | **6.06** | 87.42 |
| TinyImageNet | BadNets-white | 98.51 | 61.46 | 0.48 | 51.06 | 0.24 | 59.19 | **0.12** | **66.35** |
| | BadNets-pattern | 100.0 | 62.72 | 75.72 | 55.04 | 0.25 | 60.59 | **0.0** | **66.62** |
| | ISSBA | 99.62 | 63.1 | 87.18 | 0.66 | 0.08 | **57.56** | **0.05** | 40.6 |
| | BATT | 99.98 | 66.66 | 89.8 | 60.16 | **0.02** | 62.36 | 3.07 | **64.15** |
| | Blended | 100.0 | 70.21 | 65.22 | 61.38 | **0.0** | 64.06 | **0.0** | **66.41** |
| | Trojan-WM | 99.96 | 69.89 | 90.68 | 60.9 | 99.31 | **68.92** | **0.0** | 67.33 |
| | Trojan-SQ | 99.79 | 63.56 | 97.5 | 57.46 | 99.74 | 63.04 | **0.0** | **67.28** |
| | Smooth | 99.35 | 68.58 | 4.62 | 60.68 | **0.03** | 61.74 | 0.17 | **66.03** |
| | l0-inv | 100.0 | 63.14 | 99.76 | 54.48 | 99.99 | 44.29 | **0.0** | **65.89** |
| | l2-inv | 99.82 | 65.43 | 0.44 | 59.32 | 0.04 | 57.92 | **0.01** | **64.5** |
| | SIG | 67.99 | 71.65 | 19.04 | 63.58 | 83.67 | 59.04 | **7.77** | **71.71** |
| | **Average** | 96.82 | 66.04 | 57.31 | 53.16 | 34.85 | 59.88 | **1.02** | **64.26** |

refer to Appendix A.1 for visualisations of backdoored images. Although ViT-specific backdoor attacks exist in literature (Zheng et al., 2023; Lv et al., 2021), we did not include these attacks due to their focus on inference-time attacks. Moreover, they use threat models that are incompatible with ours. For example, Zheng et al. (2023) injects a trigger at inference-time (which does not concern finetuning), while Lv et al. (2021) modifies the finetuning procedure (which is controlled by the defender in our threat model) by using an attacker-specified loss function.

**Datasets and default parameters**. We used three datasets to evaluate our defence (CIFAR10 Krizhevsky (2009), GTSRB Houben et al. (2013), and TinyImageNet[1] Deng et al. (2009)). Defence parameters are shown in Appendix A.2.

## 4.1 Main Results

We show the results of our IEU compared to other baselines in Table 3. Our method's ASR out-performs I-BAU by 31.46 percentage points (pp) on CIFAR10 and out-performs ABL by 33.83pp on TinyImageNet. In addition, our IEU's CA for CIFAR10 and TinyImageNet are generally better than the corresponding values of the baselines. Our method has the lowest ASR in all attacks and 9 out of 11 attacks in CIFAR10 and TinyImageNet, respectively. Moreover, our method produces the highest CA for 9 out of 11 attacks for both CIFAR10 and TinyImageNet. We explore the limitations of IEU in Section 5 for weaker attacks and for the GTSRB dataset. Note that I-BAU uses the highest amount of GPU memory (39 GB on an NVIDIA A100) when compared to ABL and IEU ($\leq$ 20 GB).

## 4.2 Ablation Study on Hyperparameters

**The importance of logit masking** is shown in Table 4, which demonstrates performance degradation of IEU when logit masking is not used. Without logit masking, $f_r$ both learns and unlearns $(\mathbf{x}^{\hat{p}}, \mathbf{y}^{\hat{p}}) \in \mathcal{D}^{\mathrm{ul}}$, which by default is not learned in IEU. If the robust module performs poorly on potentially poisoned data because of asynchronous unlearning, the absence of logit masking allows the model to relearn the poisoned data during parameter updates for finetuning. Therefore, the model repeatedly learns and unlearns the same data, resulting in low performance on non-poisoned data. This reasoning also guides our decision to use $f_p$ instead of $f_r$ to isolate $\mathbf{x}^{\hat{p}}$ for Interleaved Unlearning.

Table 4: Performance of IEU without applying logit masking during finetuning. Leaving out logit masking means that $\hat{\mathbf{y}} = \hat{\mathbf{y}}_r$ is used instead of using Equation 1. All values given in percentages.

| Dataset | BATT | | BadNets-white | | ISSBA | | Smooth | |
|---------|------|------|------|------|------|------|------|------|
| | ASR | CA | ASR | CA | ASR | CA | ASR | CA |
| CIFAR10 | 0.00 | 79.75 | 0.61 | 81.49 | 0.00 | 83.42 | 0.00 | 64.06 |
| TinyImageNet | 0.36 | 56.12 | 0.01 | 28.85 | 0.00 | 11.98 | 0.00 | 26.49 |

Table 5: Performance of IEU when finetuned using $\mathcal{D}^{\mathrm{tune}}$ with different poisoning rate values using CIFAR10, $\alpha = \{0.02, 0.05, 0.1, 0.15, 0.2\}$. All values given in percentages.

| Attack | $\alpha = 0.02$ | | 0.05 | | 0.10 | | 0.15 | | 0.20 | |
|--------|------|------|------|------|------|------|------|------|------|------|
| | ASR | CA | ASR | CA | ASR | CA | ASR | CA | ASR | CA |
| BadNets-white | 1.30 | 88.27 | 1.22 | 94.05 | 0.96 | 98.19 | 0.81 | 97.14 | 1.16 | 97.85 |
| ISSBA | 0.04 | 91.78 | 0.00 | 96.78 | 0.33 | 98.35 | 0.16 | 98.15 | 0.00 | 98.10 |

**Defence performance for varying poisoning rate** is shown in Table 5. We test a wide range of poison rates to determine the effectiveness of IEU under different attack settings. Overall, our IEU is able to defend against backdoor attacks with both high and low poisoning rate. The decrease in CA when poisoning rate is low is due to the worse performance of $f_p$ at detecting potentially poisoned samples as shown in Table 1. We believe that better isolation methods for collating $\mathcal{D}^{\mathrm{ul}}$ (Doan et al., 2023) will result in higher CA.

**Effects of different confidence threshold values** are shown in Table 6. As one of the important hyperparameters in our IEU, varying $c_{\mathrm{thresh}}$ does not significantly affect model performance for both

---

[1]Used in Stanford's course CS231N. Download: `http://cs231n.stanford.edu/tiny-imagenet-200.zip`

CIFAR10 and TinyImageNet. Looking at the "Poison" and "Clean" columns of Table 6, we generally see less data (ether poisoned or clean) having maximum class probability above the confidence threshold as $c_{\text{thresh}}$ increases. Based on this observation, we argue that the performance of IEU remains stable even as $\mathcal{D}^{\text{ul}}$ decreases in size.

Table 6: Performance of IEU with $c_{\text{thresh}} \in \{0.9, 0.95, 0.99\}$. The values in the "Poison" and "Clean" columns correspond to the percentage of poisoned and clean data, respectively, that's classified as poisoned data by $f_p$ for the corresponding $c_{\text{thresh}}$. Note that $c_{\text{thresh}} = 0.95$ is the default setting. All values given in percentages.

| Dataset | $c_{\text{thresh}}$ | BATT | | | | BadNets-white | | | | Smooth | | | |
|---|---|---|---|---|---|---|---|---|---|---|---|---|---|
| | | ASR | CA | Poison | Clean | ASR | CA | Poison | Clean | ASR | CA | Poison | Clean |
| CIFAR10 | 0.90 | 0.06 | 94.42 | 94.51 | 13.61 | 0.81 | 97.47 | 95.71 | 11.53 | 0.05 | 96.79 | 97.23 | 15.16 |
| | 0.95 | 0.02 | 98.23 | 95.83 | 7.10 | 0.96 | 98.19 | 95.00 | 5.62 | 0.09 | 97.77 | 95.81 | 7.79 |
| | 0.99 | 1.74 | 98.09 | 90.68 | 0.78 | 1.27 | 98.09 | 92.70 | 0.55 | 0.32 | 97.98 | 90.76 | 1.26 |
| TinyImageNet | 0.90 | 6.73 | 65.31 | 92.68 | 0.52 | 0.12 | 65.10 | 89.45 | 0.48 | 0.00 | 66.03 | 92.27 | 0.88 |
| | 0.95 | 3.07 | 64.15 | 88.48 | 0.11 | 0.12 | 66.35 | 87.44 | 0.21 | 0.17 | 66.03 | 89.30 | 0.41 |
| | 0.99 | 1.10 | 65.41 | 92.19 | 0.00 | 0.12 | 65.35 | 82.03 | 0.01 | 2.45 | 64.65 | 80.17 | 0.07 |

**The complexity of the poisoned module** as represented by the depth of $f_p$ significantly affects the defence performance of IEU as shown in Table 7. As the depth of $f_p$ increases, it becomes more complex and more adept at learning non-poisoned samples. Since $f_p$ is confident about a larger number of clean images, this causes the number of clean data in $\mathcal{D}^{\text{ul}}$ to become higher and reduces the amount of data learned by $f_r$. Therefore, as $f_p$ becomes deeper, CA decreases since the robust module unlearns more clean data (rows for CIFAR10 and TinyImageNet of Table 7). This effect is especially pronounced for simpler datasets (e.g., CIFAR10) because simpler datasets are more easily learned given the same model complexity, leading to more clean images being directed to $\mathcal{D}^{\text{ul}}$. Tuning $c_{\text{thresh}}$ for different depth leads to better performance as shown in the last two rows of Table 7.

Table 7: Performance of IEU with varying poisoned module depth. The $c_{\text{thresh}}$ values used for the last three rows are chosen after inspecting the distribution of maximum class probability values $\max(\sigma(f_p(\mathbf{x}; \boldsymbol{\theta}_p))$ using poisoned training data. All values given in percentages.

| | Depth ($c_{\text{thresh}}$) | BadNets-white | | Blended | | ISSBA | |
|---|---|---|---|---|---|---|---|
| | | ASR | CA | ASR | CA | ASR | CA |
| CIFAR10 | 1 (0.95) | 0.96 | 98.19 | 0.00 | 98.27 | 0.33 | 98.35 |
| | 2 (0.95) | 1.30 | 56.09 | 0.00 | 86.49 | 0.00 | 87.90 |
| | 3 (0.95) | 0.00 | 18.04 | 0.00 | 64.83 | 0.00 | 34.21 |
| TinyImagenet | 1 (0.95) | 0.12 | 66.35 | 0.00 | 66.41 | 0.05 | 40.60 |
| | 2 (0.95) | 0.02 | 60.26 | 0.00 | 63.77 | 0.04 | 37.68 |
| | 3 (0.95) | 0.01 | 59.11 | 0.00 | 61.80 | 0.12 | 36.50 |
| CIFAR10 (Variable $c_{\text{thresh}}$) | 2 (0.99) | 0.92 | 97.65 | 0.00 | 92.10 | 0.00 | 98.25 |
| | 3 (0.998) | 0.82 | 92.79 | 0.00 | 98.20 | 0.00 | 98.02 |

## 4.3 ABLATION STUDY ON DEFENCE DESIGN

**We demonstrate that IEU works well for Vision Transformer variants and CNN architectures** in Table 8. We evaluate IEU where the following Vision Transformer variants are used as the robust module: CaiT-XXS (Touvron et al., 2021b), DeiT-S (Touvron et al., 2021a), PiT-XS (Heo et al., 2021), ViT-S (default architecture, Dosovitskiy et al. (2021)), and XCiT-Tiny (El-Nouby et al., 2021). In addition, we use ResNet-18 (He et al., 2015) and WideResNet-50-2 (Zagoruyko & Komodakis, 2017) to evaluate our defence on non-ViT architectures. The Interleaved Ensemble Unlearning framework generally performs well for most architectures. In addition, IEU trains high-performing models when $\alpha = 0$ where $\mathcal{D}^{\text{tune}}$ is clean (see "No Attack" column of Table 8).

**The effects of using constant unlearning rate** is shown in Table 9. Our defence is slightly more effective when $lr^{\text{ul}}$ and $lr^{\text{tune}}$ differ by a small constant factor (first three rows of CIFAR10 & TinyImageNet in Table 9). However, on average there is only a small difference between the performance of Dynamic and constant $lr^{\text{ul}}$. For example, Dynamic $lr^{\text{ul}}$ on average achieves $65.97\%$ CA on TinyImageNet, only $2.48$pp lower than the best CA at $lr^{\text{ul}} = lr^{\text{tune}}$; ASR is comparable. To have fewer hyperparameters, we use Dynamic $lr^{\text{ul}}$ instead of $lr^{\text{ul}} = c \cdot lr^{\text{tune}}$ for hyperparameter $c$.

Table 8: Performance of IEU with different model architectures using CIFAR10. The penultimate column showcases CA when applying IEU on clean $\mathcal{D}^{\text{tune}}$. The "No Defence" column uses BadNets-white as the attack and is tuned without defence. The first layer of ViT-S is used as the poisoned module for all models. We use $c_{\text{thresh}} = 0.99$ for "No Attack" since this choice mounts an effective defence as shown in Table 6. All values given in percentages.

| Variant | BATT | | BadNets-white | | ISSBA | | Smooth | | No Attack | | No Defence | |
|---|---|---|---|---|---|---|---|---|---|---|---|---|
| | ASR | CA | ASR | CA | ASR | CA | ASR | CA | ASR | CA | ASR | CA |
| CaiT-XXS | 0.01 | 97.01 | 0.98 | 97.07 | 0.76 | 97.11 | 1.20 | 97.09 | - | 97.40 | 96.80 | 97.09 |
| DeiT-S | 0.25 | 97.83 | 1.00 | 97.94 | 0.32 | 98.22 | 2.05 | 97.83 | - | 98.21 | 96.96 | 97.96 |
| PiT-XS | 0.09 | 96.44 | 1.14 | 96.62 | 4.73 | 96.74 | 3.53 | 96.66 | - | 96.78 | 96.70 | 96.66 |
| ViT-S | 0.02 | 98.14 | 0.93 | 97.95 | 0.06 | 98.15 | 0.09 | 97.61 | - | 97.87 | 97.34 | 98.12 |
| XCiT-Tiny | 0.96 | 87.72 | 0.99 | 87.67 | 1.04 | 85.47 | 4.66 | 83.96 | - | 92.48 | 95.07 | 91.79 |
| ResNet-18 | 0.35 | 90.99 | 7.22 | 91.68 | 6.37 | 91.00 | 6.38 | 91.45 | - | 92.45 | 94.25 | 91.73 |
| VGG-11 | 3.19 | 88.86 | 8.39 | 90.22 | 11.63 | 89.45 | 9.70 | 73.78 | - | 91.22 | 95.36 | 90.59 |

Table 9: Performance of IEU using different ways of computing $lr^{\text{ul}}$. Average ASR/CA across rows are shown in the last column. Dynamic $lr^{\text{ul}}$ is computed using Equation 3. All values given in percentages.

| Dataset | $lr^{\text{ul}}$ | BATT | | BadNets-white | | Smooth | | Trojan-WM | | l0-inv | | Average | |
|---|---|---|---|---|---|---|---|---|---|---|---|---|---|
| | | ASR | CA | ASR | CA | ASR | CA | ASR | CA | ASR | CA | ASR | CA |
| CIFAR10 | $1 \cdot lr^{\text{tune}}$ | 0.02 | 98.02 | 0.72 | 97.97 | 0.05 | 98.01 | 0.00 | 93.84 | 0.00 | 97.76 | 0.20 | 96.96 |
| | $2 \cdot lr^{\text{tune}}$ | 0.01 | 93.95 | 0.72 | 93.16 | 0.04 | 93.56 | 0.00 | 85.59 | 0.00 | 88.57 | 0.19 | 91.56 |
| | $4 \cdot lr^{\text{tune}}$ | 0.00 | 79.37 | 0.63 | 83.16 | 0.01 | 70.50 | 0.00 | 70.50 | 0.00 | 76.45 | 0.16 | 75.87 |
| | Dynamic | 0.02 | 98.23 | 0.96 | 98.19 | 0.09 | 97.77 | 0.00 | 98.15 | 0.00 | 98.19 | 0.27 | 98.09 |
| TinyImageNet | $1 \cdot lr^{\text{tune}}$ | 2.47 | 67.21 | 0.15 | 68.72 | 0.03 | 68.79 | 0.00 | 69.07 | 0.00 | 68.04 | 0.66 | 68.45 |
| | $2 \cdot lr^{\text{tune}}$ | 0.02 | 66.87 | 0.05 | 63.68 | 0.02 | 63.66 | 0.00 | 67.04 | 0.00 | 60.93 | 0.02 | 65.31 |
| | $4 \cdot lr^{\text{tune}}$ | 0.00 | 63.38 | 0.07 | 47.53 | 0.00 | 51.10 | 0.00 | 49.08 | 0.00 | 29.72 | 0.02 | 52.77 |
| | Dynamic | 3.07 | 64.15 | 0.12 | 66.35 | 0.17 | 66.03 | 0.00 | 67.33 | 0.00 | 65.89 | 0.84 | 65.97 |

## 5 DISCUSSION AND LIMITATIONS

The Interleaved Unlearning Framework is a high-performing defence for tuning benign models on backdoored datasets. The **impact** is that this novel framework is an **improvement for ViTs** in terms of stability and performance over existing unlearning-based methods that aim to cleanse models *after* tuning on backdoored data.

**Why do we use Local Gradient Descent (LGA) to tune** $f_p$ on the GTSRB dataset? Since the GTSRB dataset contains easily-learnt associations between benign images and their labels, the small learning capacity of $f_p$ still learns a significant amount of benign features. LGA unlearns data whose cross entropy loss is below a threshold $\gamma$. This means that benign data whose loss does not quickly decrease past an appropriately-chosen $\gamma$ will be unlearned when $\ell(\cdot, \cdot) \approx \gamma$. In contrast, the loss of backdoored data quickly decreases to around zero, where unlearning has a smaller effect due to the small magnitude of the gradient ($f_p$ overfits on poisoned images, meaning that the parameters $\boldsymbol{\theta}_p$ are close to optimal. Hence, the gradient on poisoned data is close to zero). Unlearning benign data whose loss is larger and for which $\boldsymbol{\theta}_p$ is far from optimal leads to a greater effect as the magnitude of the gradient is greater. Therefore, using LGA to tune $f_p$ causes benign data to not be learnt, thus preventing clean images from being unlearnt during interleaved unlearning. This is verified in Figure 2, where without LGA the percentage of clean data whose maximum class probability exceeds 95% is 14.6pp higher compared to tuning with LGA.

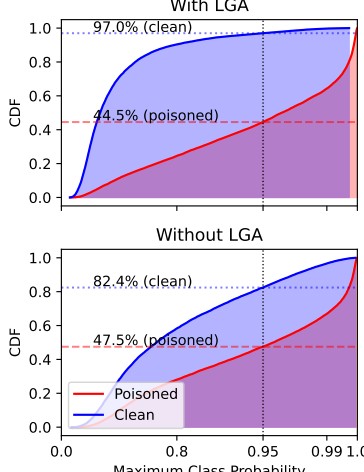

Figure 2: Maximum class probability $\max(\sigma(f_p(\mathbf{x}; \boldsymbol{\theta}_p))$ CDF based on logits produced by the poisoned module on clean and poisoned data for the ISSBA attack on GTSRB where $f_p$ is tuned with (top) and without (bottom) LGA in stage 1. Dotted horizontal lines show percentages of clean/poisoned data whose $\max(\sigma(f_p(\mathbf{x}; \boldsymbol{\theta}_p))$ lie below 0.95.

Weakness [a] **The weaker the attack, the worse the defender's performance.** We believe that defending/detecting weak attacks is as important as defending strong attacks. A key difficulty in designing performant unlearning-based backdoor defence methods is identifying and mitigating weak attacks. An example of a relatively weak attack is WaNet (Nguyen & Tran, 2021), and as shown in Table 10, the two unlearning-based methods (our IEU and ABL) we consider are less performant. Weakness [a] and [b] have a similar root cause. Both weaknesses are caused by a less effective $f_p$ for isolating backdoored data. This effect is also seen to a smaller extent against the clean label SIG attack (Barni et al., 2019) on TinyImageNet, where the ASR without defence is $\approx 68\%$. As shown in Table 3, although ASR for SIG when defended using IEU is the lowest when comparing across different defences, the ASR for SIG is higher compared to the ASR on other attacks when defending using our IEU.

**Solutions** for weakness [a]. A better isolation method can be used in place of $f_p$, such as in Doan et al. (2023). We surmise that using Doan et al. (2023)'s isolation method would make interleaved unlearning even more effective: although our IEU does not require poisoned data isolation rate to be close to 100% (see the ASR and CA values in Table 6 where adjacent "Poison" values are $\approx 80\%$ and Table 2 where $\hat{\alpha}_i \div \alpha \in \{0.5, 0.9\}$), a more effective isolation method causes the robust module to learn less backdoored data and unlearn less clean data.

Table 10: Performance of different backdoor defences on the WaNet attack (Nguyen & Tran, 2021) using the CIFAR10 dataset. Best and second-best values are bolded and underlined, respectively. All values are reported in percentages.

| No Defence | | I-BAU | | ABL | | IEU | |
|---|---|---|---|---|---|---|---|
| ASR | CA | ASR | CA | ASR | CA | ASR | CA |
| 72.81 | 97.39 | **10.26** | 90.24 | 0.00 | 10.00 | 26.01 | **94.17** |

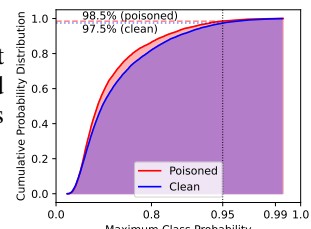

Figure 3: $\max(\sigma(f_p(\mathbf{x}\,;\boldsymbol{\theta}_p))$ CDF on clean/poisoned data for WaNet using CIFAR10.

Weakness [b] **Instability during defence** in stage 2 occurs when defending the VGG-11 model architecture against the Smooth attack. The instability causes NaN loss values during finetuning.

Potential **solutions** for weakness [b]: either (1) using $lr^{\text{ul}} = c \cdot lr^{\text{tune}}$ for hyperparameter $c$ instead of the Dynamic $lr^{\text{ul}}$ explained in Equation 3 or (2) replacing $\ell(f_r(\mathbf{x}_{k-1}^{\hat{p}}\,;\boldsymbol{\theta}_r), \mathbf{y}_{k-1})$ with a weighted moving average of successive cross entropy losses in Equation 3 may prevent instabilities from being introduced when alternating between finetuning and unlearning steps.

# 6 CONCLUSION

This work presents a novel and highly effective method for finetuning benign ViTs on backdoored datasets called Interleaved Ensemble Unlearning (IEU). We use a small and shallow ViT (the poisoned module) to distinguish between clean and backdoored images and show that alternating between learning clean data and unlearning poisoned data during defence is an effective way of preserving high clean accuracy whilst foiling the backdoor attack. We demonstrate that our defence is effective for complicated real-world datasets and discuss ways to make IEU more robust.

**Impact**. This paper's impact goes beyond developing a backdoor defense that works particularly well on ViTs. We believe that the Interleaved Unlearning framework, which extends ABL (Li et al., 2021b) and Denoised PoE (Liu et al., 2024), can be used to tune benign models with a great variety of different model architectures. In addition, we encourage future work to consider and remedy the weaknesses we point out in Section 5 for unlearning-based backdoor defences.

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

## A  MORE IMPLEMENTATION DETAILS

### A.1  BACKDOOR ATTACK DETAILS

For SIG (Barni et al., 2019) we poison 100% of the chosen target class regardless of the dataset. We mostly base our data poisoning code on BackdoorBox's attacks (Li et al., 2023c); for attacks that are not available in BackdoorBox, we adapt our code from the attack authors' source code. We modify the code for BATT (Xu et al., 2023b) implemented in BackdoorBox by moving the attacker-specified transformation to before the data augmentation step. For ISSBA (Li et al., 2021d), one encoder/decoder pair is trained for each of the three datasets. Figure 4 shows visualisations of backdoored images on CIFAR10. Throughout the paper, the target class used by the attacker is class 1.

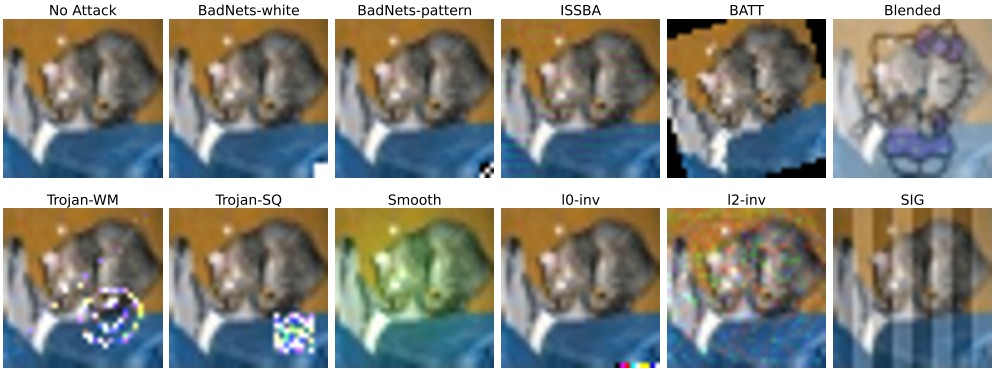

Figure 4: Visualisation of backdoored images (CIFAR10).

### A.2  BACKDOOR DEFENCE DETAILS

The data augmentations used for finetuning without defence and in our method (IEU) are based on the augmentations in Atito et al. (2021) where `drop_perc` and `drop_replace` are set to 0.3 and 0.0, respectively. Three views of each image are produced: an unaugmented image, a clean crop with only colour jitter, and a corrupted crop with colour jitter and patch-based corruption. We used the pretrained checkpoint in Atito et al. (2021) for finetuning. Note that only the non-corrupted view is used to finetune models when demonstrating the effectiveness of IEU with different model architectures in Table 8. Each image's spatial dimension is $224 \times 224$ pixels. For all experiments except for those found in Table 8, ViT-S is used as the base architecture: patch size of the ViT is $16 \times 16$; we use `embed_dim` $= 384$, `num_heads` $= 6$, `mlp_ratio` $= 4$, and LayerNorm (Ba et al., 2016).

**Finetuning without defence.** On all datasets, the ViT is finetuned for 10 epochs using the Adam (Kingma & Ba, 2015) optimizer with initial learning rate $2 \cdot 10^{-5}$ and a cosine annealing learning rate scheduler that terminates at $1 \cdot 10^{-6}$, weight decay using a cosine annealing scheduler starting from 0.04 and increasing to 0.1, and *effective* batch size 128 (recall that the two augmented views of each image is used during finetuning).

Table 11: Defence parameters used for CIFAR10/TinyImageNet and GTSRB. Parameters for CIFAR10/TinyImageNet have never been tuned (i.e., they were set to these values prior to running *any* experiments with IEU).

| Dataset | Stage 1 used LGA | Stage 1 Epochs/Warmup | Stage 1 lr | $c_{\text{thresh}}$ | lr decay |
|---|---|---|---|---|---|
| GTSRB | Yes | 10/Yes | $2 \cdot 10^{-4}$ | 0.9 | No |
| CIFAR10/TinyImageNet | No | 10/Yes | $2 \cdot 10^{-4}$ | 0.95 | No |

**Our method (IEU).** For *stage 1*, the differences in the hyperparameters used for CIFAR10/TinyImageNet and GTSRB to pre-finetune the poisoned module are shown in Table 11.

A weight decay of $0.0$ is applied to the poisoned module throughout stage 1 and no data augmentations are applied. If learning rate warmup is used, the learning rate scheduler performs a linear warmup for one epoch starting from $lr = 0$. The batch size is 64. Recall that poisoned module is a shallow ViT of depth 1. For *stage 2*, we use the same parameters during finetuning as for finetuning without defence. Please refer to Section 3 and Equation 3 for an explanation of the unlearning rate $lr^{\text{ul}}$.

**ABL.** Since the model architecture we use is different compared to the architectures used in Li et al. (2021b), we perform hyperparameter tuning using the BadNets-white attack on all three datasets. The isolation ratio $r_{\text{isol}}$ (fraction of $\mathcal{D}^{\text{tune}}$ to unlearn) is set to 0.01 and every image in $\mathcal{D}^{\text{ul}}$ is poisoned by default since we verify that LGA/Flooding can accurately select poisoned images on BadNets-white using CIFAR10. Table 12 shows the ABL hyperparameter tuning results for all three datasets. We choose $5 \cdot 10^{-7}, 1 \cdot 10^{-6}, 2 \cdot 10^{-7}$ as the unlearning rate for CIFAR10, GTSRB, and TinyImageNet, respectively. In addition, we use the Adam optimiser.

Table 12: Hyperparameter tuning for ABL on CIFAR10, GTSRB, and TinyImageNet. Bolded are reasonably good values that correspond to our choices for unlearning lr.

| Unlearning lr | CIFAR10 | | GTSRB | | TinyImageNet | |
|---|---|---|---|---|---|---|
| | ASR | CA | ASR | CA | ASR | CA |
| $5.0 \cdot 10^{-8}$ | 97.29 | 98.40 | - | - | 97.80 | 61.34 |
| $1.0 \cdot 10^{-7}$ | 97.02 | 98.41 | - | - | 95.61 | 61.06 |
| $2.0 \cdot 10^{-7}$ | 96.19 | 98.38 | 94.53 | 95.56 | **0.25** | **59.29** |
| $3.0 \cdot 10^{-7}$ | 85.64 | 98.37 | 93.85 | 95.49 | 0.03 | 57.30 |
| $5.0 \cdot 10^{-7}$ | **9.63** | **98.08** | 83.84 | 95.46 | 0.00 | 45.52 |
| $1.0 \cdot 10^{-6}$ | 7.00 | 94.73 | **4.84** | **93.56** | 0.00 | 3.06 |
| $2.0 \cdot 10^{-6}$ | - | - | 0.00 | 83.11 | - | - |
| $3.0 \cdot 10^{-6}$ | - | - | 0.00 | 72.39 | - | - |
| $5.0 \cdot 10^{-6}$ | 0.00 | 84.78 | 0.00 | 36.74 | 0.00 | 0.50 |
| $1.0 \cdot 10^{-5}$ | 0.00 | 10.00 | 0.00 | 3.56 | 0.00 | 0.50 |
| $5.0 \cdot 10^{-5}$ | 0.00 | 10.00 | 0.00 | 3.56 | 0.00 | 0.50 |
| $1.0 \cdot 10^{-4}$ | 0.00 | 10.00 | 0.00 | 0.95 | 0.00 | 0.50 |

**I-BAU.** We also perform hyperparameter tuning for I-BAU (Zeng et al., 2021b) for the same reasons above. Following suggestions in the appendix of Wang et al. (2022a), we tune `outer_lr` $\in \{5 \cdot 10^{-4}, 1 \cdot 10^{-4}, 5 \cdot 10^{-5}, 1 \cdot 10^{-5}, 5 \cdot 10^{-6}\}$ and `inner_lr` $\in \{0.1, 1, 5, 10, 20\}$ for CIFAR10 and TinyImageNet. GTSRB is not separately tuned since good eprformance is reached using CIFAR10's hyperparameters. We choose as the hyperparameters $(\texttt{outer\_lr}, \texttt{inner\_lr}) = [(5 \cdot 10^{-5}, 5), (5 \cdot 10^{-5}, 5), (5 \cdot 10^{-5}, 10)]$ for CIFAR10, GTSRB, and TinyImageNet, respectively. In addition, we use the Adam optimiser as the outer optimiser for I-BAU. For every dataset, 5000 images from the *testing set* are used for the unlearning step (`unlloader` in their code). Note that 5000 clean images taken from the *testing set* is the default setup in the defence code of I-BAU.

**AttnBlock.** This is the defence referred to as the "*Attn Blocking*" defence in Subramanya et al. (2024). We use GradRollout (Gildenblat, 2020) to compute the interpretation map on image $\mathbf{x}$ $\mathbf{I}_{\text{map}}(\mathbf{x})$ using the backdoored checkpoint and find the coordinates of the interpretation map's maximum $\max(\mathbf{I}_{\text{map}}(\mathbf{x}))$. A $30 \times 30$ patch centred at the coordinates $\max(\mathbf{I}_{\text{map}})$ is zeroed out from the original image $\mathbf{x}$ to form $\mathbf{x}'$. If this centred patch goes outside of the image, the patch is shifted so that it is on the image's border. Then, the backdoored checkpoint is used to classify $\mathbf{x}'$. The results for the test-time interpretation-informed defence proposed in Subramanya et al. (2024) is shown in Table 13. Generally, the defence does not defend against the non-patch-based attacks evaluated in our work.

## A.3 HARDWARE RESOURCES

Most experiments are conducted on one NVIDIA A100 GPU. A few experiments are (and can be) conducted on one NVIDIA Quadro RTX 6000 GPU. We did not reproduce I-BAU (Zeng et al., 2021b) on the RTX 6000 GPU due to GPU memory constraints.

Table 13: Performance of AttnBlock

| Attack | CIFAR10 | | GTSRB | | TinyImageNet | |
|---|---|---|---|---|---|---|
| | ASR | CA | ASR | CA | ASR | CA |
| BadNets-white | 34.71 | 98.21 | 29.05 | 94.73 | 54.35 | 60.21 |
| BadNets-pattern | 12.61 | 98.06 | 22.03 | 94.2 | 7.55 | 61.2 |
| ISSBA | 100.0 | 97.93 | 100.0 | 94.13 | 99.27 | 61.82 |
| BATT | 99.99 | 98.0 | 100.0 | 94.63 | 99.99 | 65.42 |
| Blended | 99.99 | 98.15 | 100.0 | 94.8 | 100.0 | 68.94 |
| Trojan-WM | 99.98 | 98.13 | 99.98 | 91.35 | 99.97 | 68.82 |
| Trojan-SQ | 99.48 | 98.09 | 99.47 | 94.06 | 99.63 | 61.97 |
| Smooth | 99.72 | 97.99 | 99.71 | 94.66 | 99.17 | 67.54 |
| l0-inv | 100.0 | 98.1 | 100.0 | 95.14 | 100.0 | 62.1 |
| l2-inv | 99.99 | 98.2 | 100.0 | 92.77 | 99.78 | 64.33 |
| SIG | 98.35 | 88.53 | 99.43 | 90.42 | 68.36 | 70.41 |
| **Average** | 85.89 | 97.22 | 86.33 | 93.72 | 84.37 | 64.8 |

## B  ALGORITHM FOR STAGE 2

The steps for Stage 2 of IEU is shown in Algorithm 1.

---

**Algorithm 1** Stage 2 of IEU: Defending $f_r$

---

1: **Input**: tuned poisoned module $f_p(\cdot\,;\boldsymbol{\theta}_p)$, potentially poisoned finetuning set $\mathcal{D}^{\text{tune}}$, pretrained benign $f_r(\cdot\,;\boldsymbol{\theta}_r)$, optimizer for finetuning $\text{Optim}^{\text{tune}}$, optimizer for unlearning $\text{Optim}^{\text{ul}}$, number of epochs, and learning rate schedule $lr^{\text{tune}}$.

2: **Output:** tuned robust module $f_r(\cdot\,;\boldsymbol{\theta}_r)$ that is benign.

3: **for** every epoch **do**

4:     Initialise $\mathcal{D}^{\text{ul}}$ to be an empty queue to store potentially poisoned images.

5:     **for** every batch $(\mathbf{x}_b, \mathbf{y}_b) \in \mathcal{D}^{\text{tune}}$ **do**                    ▷ The subscript 'b' indicates a batch.

6:         Compute $\hat{\mathbf{y}}_b$ using equation 1, $f_p$, and $f_r$ for this batch.

7:         Compute the cross entropy loss $\ell(\hat{\mathbf{y}}_b, \mathbf{y}_b)$.

8:         Update $\boldsymbol{\theta}_r$ using $\text{Optim}^{\text{tune}}$ to optimise for the "Learn" section of Equation 2.

9:         Update $lr^{\text{tune}}$ based on the learning rate schedule.

10:         Add all $(\mathbf{x}_{b,i}, \mathbf{y}_{b,i}) \in (\mathbf{x}_b, \mathbf{y}_b)$ that satisfies $\max(\sigma(f_p(\mathbf{x}_{b,i}\,;\boldsymbol{\theta}_p))) > c_{\text{thresh}}$ to $\mathcal{D}^{\text{ul}}$.

11:         **if** a batch $(\mathbf{x}_b^{\hat{p}}, \mathbf{y}_b^{\hat{p}}) \in \mathcal{D}^{\text{ul}}$ is ready **then**

12:             Compute $\ell(f_r(\mathbf{x}_b^{\hat{p}}\,;\boldsymbol{\theta}_r), \mathbf{y}_b^{\hat{p}})$ and $lr^{\text{ul}}$ based on Equation 3.

13:             Update $\boldsymbol{\theta}_r$ using $\text{Optim}^{\text{ul}}$ to optimise for the "Unlearn" section of Equation 2 and record $lr^{\text{ul}}$ for the next batch.

14:             Dequeue the current batch $(\mathbf{x}_b^{\hat{p}}, \mathbf{y}_b^{\hat{p}})$ from $\mathcal{D}^{\text{ul}}$.

15:         **end if**

16:     **end for**

17: **end for**

---

# C  ABLATION STUDY (CONT'D)

**Noisy logits**. Table 14 shows the effects of adding normally distributed zero-mean noise to $\hat{\mathbf{y}}_p$ when accumulating the unlearn set. Whether noise is added or not has almost no effect on the model performance.

Table 14: Performance of IEU on CIFAR10 when adding normally distributed zero-mean noise with different variances to $\hat{\mathbf{y}}_p$ *after* computing $m_{\boldsymbol{\theta}_p}$, i.e., $m_{\boldsymbol{\theta}_p}$ is still defined according to Equation 1 but $\max(\sigma(\hat{\mathbf{y}}_p + \mathbf{n})) > c_{\text{thresh}}$ where $\mathbf{n} \sim \mathcal{N}(0, \sigma^2)$ is used to determine whether $\mathbf{x}$ belongs in $\mathcal{D}^{\text{ul}}$. This creates a mismatch between the data added onto $\mathcal{D}^{\text{ul}}$ (unlearned by $f_r$) and the data learned by $f_r$.

| Variance | BATT | | BadNets-white | | ISSBA | | Smooth | |
|---|---|---|---|---|---|---|---|---|
| | ASR | CA | ASR | CA | ASR | CA | ASR | CA |
| 0.0 | 0.02 | 98.23 | 0.96 | 98.19 | 0.33 | 98.35 | 0.09 | 97.77 |
| 0.1 | 0.02 | 98.09 | 0.93 | 98.23 | 0.10 | 98.43 | 0.08 | 97.85 |
| 0.5 | 0.02 | 98.21 | 0.88 | 98.13 | 0.35 | 98.36 | 0.04 | 97.64 |
| 1.0 | 0.01 | 98.15 | 0.95 | 98.09 | 0.05 | 98.30 | 0.05 | 97.91 |
| 2.0 | 0.02 | 96.82 | 0.94 | 98.17 | 0.46 | 98.30 | 0.06 | 97.95 |

**The effect of using LGA/Flooding in conjunction with our poisoned module** is shown in Table 15. The short-hand $f_p$ & $M$ means applying method $M$ when tuning $f_p$ during stage 1 of our method. Given the high FNR for most attacks with CIFAR10 and TinyImageNet when using LGA/Flooding together with our $f_p$, we argue that using poisoned module is orthogonal to LGA/Flooding for isolating poisoned data on these two datasets. However, we find that $f_p$ alone is not enough to defend $f_r$ when using the GTSRB dataset. The simplicity of the GTSRB dataset explains why LGA/Flooding is necessary to isolate poisoned data successfully: a simpler dataset leads to $f_p$ to learn the benign features more quickly, but at a slower pace compared to backdoored images. This causes the $f_p$ to be insufficient in detecting backdoored images. Therefore, LGA/Flooding is used to ensure a large gap between the poisoned module's confidence on backdoored and benign data.

Table 15: Comparison of the three configurations' (LGA, Flooding, neither method) ability to distinguish between poisoned and clean data when used in conjunction with our $f_p$ during prefinetuning. The $f_p$ is prefinetuned using default parameters (Table 11). The flooding/LGA parameter is set to $\gamma = 1.5, \gamma = 1.0, \gamma = 3.0$ for CIFRA10, GTSRB, and TinyImageNet, respectively. Each cell shows the FPR/FNR values as percentages (positive means "poisoned").

| Method | CIFAR10 | | | TinyImageNet | | | GTSRB | | |
|---|---|---|---|---|---|---|---|---|---|
| | BadNets-white | ISSBA | Smooth | BadNets-white | ISSBA | Smooth | BadNets-white | ISSBA | Smooth |
| $f_p$ & Flooding | 0.00/99.98 | 0.00/57.76 | 0.00/99.54 | 0.03/20.07 | 0.01/64.66 | 0.12/15.89 | 11.51/22.07 | 8.38/34.89 | 10.32/18.69 |
| $f_p$ & LGA | 0.00/99.98 | 0.00/65.84 | 0.00/97.86 | 0.03/20.07 | 0.12/67.32 | 0.12/15.89 | 11.51/22.07 | 8.38/34.89 | 10.32/18.69 |
| $f_p$ & Neither | 5.62/5.30 | 5.58/2.10 | 7.90/3.14 | 0.21/11.97 | 0.15/22.59 | 0.42/9.83 | 44.72/14.86 | 42.63/5.71 | 44.36/3.45 |

We note the detailed settings of stage 1 for the data presented in Table 15 here. Experiments for CIFAR10, TinyImageNet, and GTSRB with LGA/Flooding use the default parameters presented in Table 11. The stage 1 settings used for GTSRB with $f_p$ & Neither method (last row) are different: we use 5 epochs and no learning rate warmup during stage 1 pre-finetuning, $1 \cdot 10^{-3}$ as the stage 1 learning rate, $c_{\text{thresh}} = 0.998$. We tune on BadNets-white the parameters for GTSRB where only $f_p$ is used (no LGA/Flooding) in Stage 1 to reach an acceptably low FNR and FPR. Note that the stage 1 poisoned module in Table 16 for the "$f_p$ Only" column is tuned using the same settings as $f_p$ & Neither as explained above. The $f_p$ & LGA column in Table 16 follows default parameters.

**The improvement of IEU when using LGA (see Li et al. (2021b)) during Stage 1 for GTSRB** is shown in Table 16. Generally, the ASR with LGA is marginally higher than without LGA except for ISSBA, where using LGA improves the ASR by almost 100pp. This marignal increase in ASR is accompanied by a significant increase in CA compared to tuning the poisoned module without LGA.

Our IEU defence successfully defends when **using different attacker-specified trigger labels** as shown in Table 17.

Table 16: Comparison of performance of IEU on GTSRB with and without LGA during Stage 1. Note that $c_{\text{thresh}} = 0.9$ when using LGA ($\gamma = 1$) and $c_{\text{thresh}} = 0.95$ without LGA. All values given in percentages.

| Attack | $f_p$ & LGA | | $f_p$ Only | | Difference | |
|---|---|---|---|---|---|---|
| | ASR | CA | ASR | CA | ASR | CA |
| BadNets-white | 2.22 | 83.26 | 0.93 | 82.07 | +1.29 | +1.19 |
| BadNets-pattern | 0.00 | 95.11 | 0.00 | 88.53 | 0.00 | +6.58 |
| ISSBA | 2.81 | 86.48 | 100.00 | 93.52 | -97.19 | -7.04 |
| BATT | 7.40 | 88.38 | 0.03 | 93.47 | +7.37 | -5.09 |
| Blended | 6.83 | 89.25 | 8.76 | 81.20 | -1.93 | +8.05 |
| Trojan-WM | 8.87 | 91.81 | 3.12 | 86.54 | +5.75 | +5.27 |
| Trojan-SQ | 2.85 | 89.02 | 0.07 | 77.29 | +2.78 | +11.73 |
| Smooth | 15.37 | 88.45 | 6.15 | 84.52 | +9.22 | +3.93 |
| l0-inv | 0.00 | 88.81 | 0.00 | 87.48 | 0.00 | +1.33 |
| l2-inv | 20.26 | 83.67 | 12.85 | 87.50 | +7.41 | -3.83 |
| SIG | 0.00 | 77.39 | 0.21 | 72.07 | -0.21 | +5.32 |

Table 17: Effectiveness of IEU when faced with different trigger labels $\in \{0, 1, 3, 5, 8\}$ when trained and evaluated on the CIFAR10 dataset using three attacks. Note that we use class label 1 as the default target label for every experiment other than those found in this table. All values given in percentages.

| Target Label | BadNets-white | | ISSBA | | Smooth | |
|---|---|---|---|---|---|---|
| | ASR | CA | ASR | CA | ASR | CA |
| 0 | 0.71 | 92.28 | 0.10 | 98.43 | 0.07 | 97.71 |
| 1 | 0.96 | 98.19 | 0.33 | 98.35 | 0.09 | 97.77 |
| 3 | 0.51 | 97.88 | 0.03 | 98.36 | 0.10 | 93.87 |
| 5 | 0.37 | 97.53 | 0.00 | 98.35 | 0.05 | 97.71 |
| 8 | 0.19 | 94.51 | 0.00 | 98.14 | 0.01 | 97.51 |

## D  POTENTIAL ADAPTIVE ATTACK

This section investigates how IEU performs when faced with a backdoor attack that is designed to bypass our IEU. This setting is more challenging for defenders, since the attacker can take countermeasures that are specifically designed to evade IEU. Our defense mechanism uses the poisoned module to filter out highly confident data which are then classified as backdoored data. A potential attack would be one that makes the backdoor trigger more hidden to induce the stage 1 poisoned module to make incorrect classifications.

A previous work (Mo et al., 2024) designed such an attack, named "Channel Activation Attack" (abbreviated CAT), whose aim is to produce adversarial perturbations to encourage the channel activation patterns of benign and backdoored images to appear more similar. This serves as an adaptive attack against our defence. We use a gray-box setting, where the attacker has full access to a version of the poisoned module that the defender has tuned using the same exact setting as the defender would use during normal application of IEU. With a tuned version of $f_p$, the attacker adds adversarial perturbations onto existing backdoor attacks to evade detection. Using the adversarially perturbed backdoor data, we apply our IEU framework to defend the robust module. We apply the CAT attack on five different standard backdoor attacks. The results are shown in Table 18, which demonstrates that our method successfully defends the CAT attack.

We use a very similar attack setup as attack's original authors in Mo et al. (2024) ($\gamma = 0.6$ for the loss function in their Equation 3; 10 iterations and an $\ell_2$-norm budget of $\epsilon = 16 \div 255$ for the Projected Gradient Descent attack). However, we did not apply random masking to the perturbations. Please refer to Figure 5 to visualise the original backdoored image, the CAT-perturbed backdoor image, and the difference between the two.

Table 18: IEU's performance against the CAT attack proposed in Mo et al. (2024) for both CIFAR10 and TinyImageNet.

| Dataset | Attack | IEU | | No Defense | |
|---|---|---|---|---|---|
| | | ASR | CA | ASR | CA |
| CIFAR10 | CAT+BadNets-pattern | 0.00 | 98.27 | 100.00 | 97.77 |
| | CAT+BadNets-white | 0.74 | 94.90 | 94.72 | 98.31 |
| | CAT+Blended | 0.00 | 98.28 | 99.96 | 98.38 |
| | CAT+Smooth | 0.10 | 97.87 | 98.88 | 98.33 |
| | CAT+Trojan-SQ | 0.02 | 98.25 | 99.66 | 98.44 |
| TinyImageNet | CAT+BadNets-pattern | 0.00 | 68.32 | 100.00 | 69.77 |
| | CAT+BadNets-white | 0.04 | 69.43 | 95.96 | 70.09 |
| | CAT+Blended | 0.00 | 65.58 | 99.97 | 70.46 |
| | CAT+Smooth | 0.01 | 68.10 | 92.47 | 70.34 |
| | CAT+Trojan-SQ | 0.00 | 66.66 | 99.63 | 70.71 |

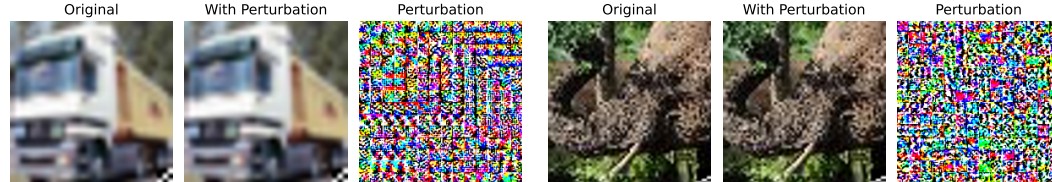

Figure 5: CAT attack example backdoor images, perturbed backdoor images, and adversarial perturbations (scaled by 5000). Left three: CIFAR10; right three: TinyImageNet.

# E   LIMITATIONS (CONT'D)

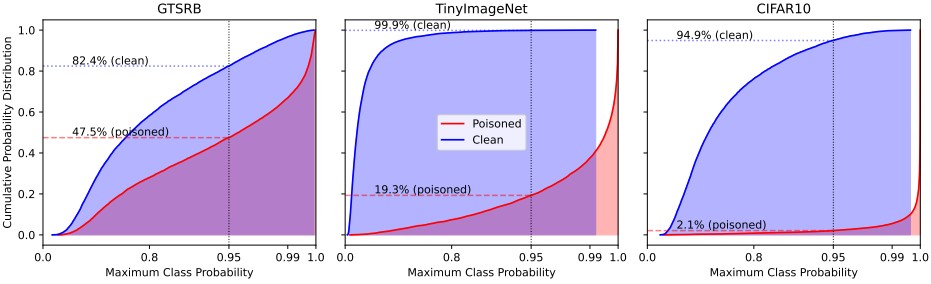

Figure 6: Maximum class probability $\max(\sigma(f_p(\mathbf{x}\,;\boldsymbol{\theta}_p)))$ CDF based on logits produced by the poisoned module on clean and poisoned data for the ISSBA attack on the three datasets. Dotted horizontal lines show percentages of clean/poisoned data whose $\max(\sigma(f_p(\mathbf{x}\,;\boldsymbol{\theta}_p)))$ lie below 0.95.

Weakness [c] **Worse performance on less complex datasets** (e.g., GTSRB). Our IEU fails against ISSBA on the GTSRB dataset when tuning $f_p$ without LGA (shown in Table 15) and underperforms on GTSRB in general. We suggest that this happens because the GTSRB dataset is easily learnt by $f_p$. Evidence is shown in Figure 6, which plots the CDF of the maximum class probability values for poisoned/clean data for all three datasets using the ISSBA attack where LGA is not used when tuning $f_p$ in stage 1. Compared to CIFAR10 and TinyImageNet, clean images from the GTSRB dataset is only marginally more difficult to learn than backdoored GTSRB images. In Figure 6, a higher proportion of clean data has $\max(\sigma(f_p(\mathbf{x}\,;\boldsymbol{\theta}_p))) > 0.95$ and a lower proportion of poisoned data has $\max(\sigma(f_p(\mathbf{x}\,;\boldsymbol{\theta}_p))) > 0.95$ for GTSRB when compared to the other two datasets. Therefore, using a shallow ViT as the poisoned module is insufficient for discerning poisoned data from clean data for GTSRB. This led us to use LGA when tuning $f_p$ when using the GTSRB dataset.

**Solutions** for weakness [c]: please refer to solutions for weakness [a] in Section 5. Additionally, using LGA also solves this problem.

