# OpenReview forum: "Using Interleaved Ensemble Unlearning to Keep Backdoors at Bay for Finetuning Vision Transformers"
_ICLR.cc/2025/Conference — ICLR 2025 Conference Withdrawn Submission_

### Official Review · Reviewer_NEex · 2024-10-28

**Soundness:** 3
**Presentation:** 3
**Contribution:** 3
**Rating:** 6
**Confidence:** 3

**Summary:**

Vision Transformers (ViTs) are increasingly used in computer vision tasks but are vulnerable to backdoor attacks that can compromise their performance, especially in security-sensitive applications. Existing backdoor defenses for Convolutional Neural Networks (CNNs) are not as effective for ViTs, and tailored solutions are limited.

To address this, the paper proposes Interleaved Ensemble Unlearning (IEU), a method for fine-tuning clean ViTs on backdoored datasets. In the first stage, a shallow ViT is fine-tuned to exhibit high confidence on backdoored data while maintaining low confidence on clean data. In the second stage, this shallow ViT serves as a "gate" to filter out potentially poisoned data from the defended ViT, which is then added to an unlearn set and asynchronously unlearned via gradient ascent.

The paper demonstrates IEU's effectiveness across three datasets against 11 state-of-the-art backdoor attacks and highlight its versatility by applying it to various model architectures.

**Strengths:**

- This paper explores a method for finetuning clean ViTs on backdoored datasets, Interleaved Ensemble Unlearning (IEU), giving a good reference to the research on this aspect.

- The proposed method IEU is simple but effective to achieve the defense, and the good performance obtained by the experiments strongly supports this point.

- The ablation study is organized well to clearly demonstrate the whole proposed method. And it makes the paper easy to follow.

**Weaknesses:**

- I have some concerns regarding **the fairness of comparison with previous defense methods,** particularly due to the inclusion of the additional poisoned module, which involves an extra network for defense.

- The authors propose an effective framework of backdoor defense on ViT. It would be beneficial to consider a more realistic scenerio, where the attacker knows the existence of IEU and can generate a poisoned dataset to perform adaptive attacks. **It would be wonderful if the authors can design an adaptive attack for IEU and provide some experimental results.**

- I recommend conducting further experiments to assess whether BSD can successfully defend against backdoor attacks **with different target labels.** This could provide valuable insights into the robustness of the defense mechanism across various scenarios.

- There are existing backdoor defenses that focus on training clean models using poisoned datasets [1, 2], which could provide useful context for this research.

[1] Backdoor Defense via Adaptively Splitting Poisoned Dataset. CVPR, 2023.

[2] Backdoor Defense via Deconfounded Representation Learning. CVPR, 2023.

**Questions:**

Listed in the weakness of the paper.

Score can be improved if concerns listed above are resolved.

---

> ### Author Response · Authors · 2024-11-13
> **Response to reviewer NEex**
>
> - **Reviewer's comment**: I have some concerns regarding the fairness of comparison with previous defense methods, particularly due to the inclusion of the additional poisoned module, which involves an extra network for defense.
>   - **Author's response**: I was inspired by DenoisedPoE [1] (a defence for backdoor attacks in language tasks). They similarly used a poisoned module (the extra network) to “capture” backdoored samples. I agree that my IEU presents an extra layer of complexity compared to other modules; since existing defences for image classification did not contain an auxiliary network in their frameworks and this poisoned module can be viewed as an integral part of my method (similar to how the unlearning stage is an integral part of ABL’s method [2]), I did not consider the fairness of my comparisons. Other papers (e.g., Trap-and-replace [3]) replace parts of their networks during their defence, so I didn’t see an issue with making comparisons with SOTA methods given that the poisoned module is an integral part of my design. I am not entirely sure how I can make comparisons with other defences in a way that’s fair, but I’m open to suggestions on how I can improve the comparisons!
> - **Reviewer's comment**: The authors propose an effective framework of backdoor defense on ViT. It would be beneficial to consider a more realistic scenario, where the attacker knows the existence of IEU and can generate a poisoned dataset to perform adaptive attacks. It would be wonderful if the authors can design an adaptive attack for IEU and provide some experimental results.
>   - **Author's response**: A known issue of my IEU is its subpar performance against weak attacks such as WaNet and easy-to-learn datasets such as GTSRB (see section 5 <= I hope you will appreciate the effort I put into explore IEU’s weaknesses. I could’ve simply avoided showing results for the WaNet attack or the GTSRB dataset, but felt that it’s important that I explore various weaknesses in my method). In light of section 5, if you would still like to see an adaptive attack, I will definitely produce one. Thank you for the suggestion!
> - **Reviewer's comment**: I recommend conducting further experiments to assess whether BSD (backdoor sample detection) can successfully defend against backdoor attacks with different target labels. This could provide valuable insights into the robustness of the defense mechanism across various scenarios.
>   - **Author's response**: I will definitely try with different target labels since I only considered target label == 1 in my original set of experiments. Thank you for the valuable suggestion!
> - **Reviewer's comment**: There are existing backdoor defenses that focus on training clean models using poisoned datasets [1, 2], which could provide useful context for this research.
>   - **Author's response**: thank you for suggesting the papers! They will be mentioned in the paper. I seem to have missed the first paper but did came upon the second one (which I probably forgot to cite).
>
> I am very happy to read your very valuable feedback! I will try my absolute best to make sure that new experimental results are presented by the end of the discussion period.
>
> [1] From Shortcuts to Triggers: Backdoor Defense with Denoised PoE
>
> [2] Anti-Backdoor Learning: Training Clean Models on Poisoned Data
>
> [3] Trap and Replace: Defending Backdoor Attacks by Trapping Them into an Easy-to-Replace Subnetwork

---

> ### Author Response · Authors · 2024-11-16
> **Results with different target labels have been posted!**
>
> Please see my comment to everyone.

---

> ### Author Response · Authors · 2024-11-20
> **IEU's performance against an adaptive attack has been posted!**
>
> Please see my comment to everyone.
>
> I have provided all additional experimental results that you have suggested. Please stay tuned for a revised version of my manuscript.
>
> Thank you so much for your comments!

---

> ### Author Response · Authors · 2024-11-23
> **Looking forward to your response!**
>
> Dear Reviewer NEex,
>
> I have addressed your concerns by providing extensive experimental data. Would it be possible for you to consider updating your score to reflect my responses or update the thread with lingering questions?
>
> Thank you!

---

> > ### Comment · Reviewer_NEex · 2024-11-30
> >
> > Thank you for your detailed feedback. For adaptive attacks, the authors provide an existing backdoor attack from [1]. However, I am specifically interested in an adaptive attack that directly targets your proposed IEU. For instance, if attackers are aware of the additional poisoned module introduced by your method and intentionally craft a poisoned dataset designed to bypass it, would the IEU still be able to defend against such an attack? I appreciate the comprehensive experiments provided for [1]. However, I believe incorporating an adaptive attack more related with your IEU would offer valuable insights into the robustness of IEU and provide readers with a deeper understanding of its effectiveness.
> >
> > [1] Towards Reliable Backdoor Attacks on Vision Transformers.

---

> ### Author Response · Authors · 2024-11-30
> **The new attack does exactly what you've requested! So, it's an adaptive attack**
>
> Thank you so much for your additional comments!
>
> The attack in [1] **does "intentionally craft a poisoned dataset designed to bypass [the poisoned module]"**. I implemented the attack in [1] because it is specifically (and serendipitously) designed to evade detection by the poisoned module, meaning that the **attack in [1] directly targets IEU** under a grey-box setting where the attack is **fully aware of the poisoned module** and can have full access to any copy of the poisoned module. This is explained in Appendix D of the revised manuscript.
>
> Here' a summary of how I implemented their attack to target the poisoned module, which hopefully will convince you that it is indeed an adaptive attack that specifically tries to defeat IEU:
> - The attacker starts off with a poisoned module that's tuned using stage 1 of IEU using a poisoned dataset that's been poisoned with an attacker-specified trigger (e.g., if the attacker wishes to use BadNets as a starting point for attacking my poisoned module, they first procure a poisoned module that's trained on BadNets)
>   - note that the attacker has access to a copy of the requested poisoned module that's tuned on the requested backdoor attack using the same exact procedure as stage 1 of IEU. Further, note that the defender will **not** be using this exact copy of the poisoned module that the attacker has access to, since the threat model used by IEU specifies that only the training data is controlled by the attacker. This means that the attacker cannot specify a certain poisoned module to be used during defence with IEU. Hence, it's a gray-box attack.
> - Then, with this poisoned module, the attacker performs the projected gradient descent attack to make backdoored images appear more hidden to the poisoned module. This is achieved by adding adversarial perturbations to the vanilla backdoored images such that the copy of the poisoned module accessed by the attacker classifies the perturbed poisoned image $x + \delta + \delta'$ (where $\delta$ is the original backdoor trigger, e.g. the BadNets trigger, and $\delta'$ is the adversarial perturbation) as a benign image.
> - The dataset with more evasive poisoned images is produced by performing projected gradient descent on all backdoored images, where the goal is to evade the poisoned module.
> - The defender then applies IEU (including stage 1) on the poisoned dataset that's specifically designed to avoid detection by the poisoned module. Note that the defender trains a new poisoned module based on this stealthy backdoored dataset.
> - Finally, backdoored images that do not have adversarial perturbations are used to test the ASR of the defended model. This is the same setting used in [1].
>
> I hope this clears things up! Please feel free to follow up with further comments or questions. I explain the backdoor attack in [1] in Appendix D of the revised manuscript. Additionally, you can look at my comment titled "IEU successfully defends against an adaptive attack" if you find it helpful.
>
> [1] Towards Reliable Backdoor Attacks on Vision Transformers.

---

> > ### Comment · Reviewer_NEex · 2024-11-30
> >
> > Thanks for your detailed feedback and clarity. I think that you can add this more clear explanation for your adaptive attack in your revised paper. I have raised my score.

---

> ### Author Response · Authors · 2024-11-30
> **Thank you for the suggestion!**
>
> I will make sure to explain in detail why the attack is indeed an adaptive attack in my paper if I get the chance to do so on Openreview... (If not, then the update will be reflected in my own copy). Apologies for the oversight.
>
> Thank you so much for time and effort you've put into the review!

---

### Official Review · Reviewer_JxzC · 2024-10-28

**Soundness:** 2
**Presentation:** 1
**Contribution:** 2
**Rating:** 3
**Confidence:** 3

**Summary:**

The paper presents Interleaved Ensemble Unlearning (IEU), a method aimed at defending Vision Transformers (ViTs) from backdoor attacks during fine-tuning on backdoored datasets. IEU employs a two-stage approach, using a shallow "poisoned module" ViT to filter potentially poisoned data and a "robust module" to learn clean data. In stage one, the poisoned module is trained to confidently predict backdoor-labeled data, while stage two involves using this module to identify and remove potentially backdoored data. This dynamic unlearning technique is tested on multiple datasets and against various attacks, showcasing IEU’s ability to improve attack success rate (ASR) and clean accuracy (CA).

**Strengths:**

1. The paper proposes a unique ensemble-based strategy specifically tailored for ViTs, addressing the scarcity of ViT-targeted backdoor defenses.

2. IEU is evaluated on diverse datasets (CIFAR10, GTSRB, TinyImageNet) and multiple backdoor attacks, providing extensive empirical data on performance improvements.

**Weaknesses:**

1. The presentation quality falls significantly short of the standards expected at a prestigious conference like ICLR. Readers without a strong background in vision transformers and backdoor attacks will likely struggle to follow the manuscript. Key background information, such as the concept of "unlearning", is notably absent. Additionally, Figure 1 presents mathematical notations to illustrate the proposed method, yet these notations lack sufficient explanation. For example, the symbol $\mathcal{D}^{ul}$ is used without clarifying its meaning.

2. Although the paper claims that IEU enhances ViT robustness against backdoor attacks, it lacks clarity on which specific design elements are tailored to address vulnerabilities unique to the ViT architecture.

3. This paper lacks novelty. The ensemble strategy primarily relies on heuristic thresholds rather than introducing innovative designs. Additionally, Stage 1 does not present any novel contributions.

**Questions:**

1. The proposed method heavily relies on the prediction of the poisoned module. What if the poisoned module itself is attacked by malicious users? How can the proposed method address this scenario?

2. The proposed method appears to be sensitive to the heuristic threshold $c_{thresh}$. How do the authors determine the value of the threshold?

---

> ### Author Response · Authors · 2024-11-13
> **Response to reviewer JxzC**
>
> - **Reviewer's comment**: Readers without a strong background in vision transformers and backdoor attacks will likely struggle to follow the manuscript. Key background information, such as the concept of "unlearning", is notably absent.
>   - **Author's response**: I modelled my paper's structure after Trap-and-Replace [1], which does not include much background information on CNNs or other model architectures. This led me to not focus on discussing Vision Transformers because, similar to CNN, ViT is another standard network architecture in deep learning. I will improve the wording of my brief explanation of backdoor attacks in the first paragraph of my introduction. In my literature review, I discussed many backdoor defenses and attacks including ViT-specific backdoor attacks, which should provide enough background for someone who’s familiar with the field of backdoor defense & attack. Other reviewers did not seem to note significant issues with my presentation. I agree that I should dedicate some parts of my related work and method sections to introducing the concept of unlearning and will update my manuscript to that effect.
> - **Reviewer's comment**: Additionally, Figure 1 presents mathematical notations to illustrate the proposed method, yet these notations lack sufficient explanation. For example, the symbol $D^{ul}$ is used without clarifying its meaning.
>   - **Author's response**: I agree that readers may find symbols confusing if there're not explained in the vicinity of the figure. In the updated manuscript I will clarify important symbols and substitute unnecessary ones with text; additionally, I will point the reader towards the start of section 3 which contains a summary of notations after I’ve explained an overview of my framework.
> - **Reviewer's comment**: Although the paper claims that IEU enhances ViT robustness against backdoor attacks, it lacks clarity on which specific design elements are tailored to address vulnerabilities unique to the ViT architecture.
>   - **Author's response**: I argue that IEU enhances ViT robustness against backdoor attacks because IEU's performance on defending ViTs is much **better compared to SOTA methods** that were previously applied on CNNs, which is a solid argument for why IEU is more applicable to ViTs. I agree with you that I did not exploit ViTs’ design more convincingly when designing IEU. My specific implementation of IEU is less appropriate for CNNs since the poisoned module is directly duplicated from a ViT layer in the defended ViT (although some parameters are duplicated, note that $f_p$ and $f_r$ do not share parameter updates during tuning). However, the concept of interleaved unlearning, which is my main contribution, is indeed not specific to addressing vulnerabilities in the ViT architecture. I will highlight the experimental results of IEU when defending ViTs to support my argument that IEU is for ViTs.
>
> - **Reviewer's comment**: This paper lacks novelty. The ensemble strategy primarily relies on heuristic thresholds rather than introducing innovative designs. Additionally, Stage 1 does not present any novel contributions.
>   - **Author's response**: I will emphasise my most important contribution more appropriately in the main text. The process of interleaved unlearning makes a novel and impactful contribution since previous SOTA unlearning-based methods do not perform well when applied to ViTs. As presented in my main results (table 3), benign features are forgotten after unlearning (see results for ABL [2] in table 3) and, often, unlearning a small subset of backdoored samples is not sufficient to erase the more pernicious and hidden backdoor triggers. I agree that stage 1 does not present any novel contributions and stage 1 is not the focus of the paper. It is simply a preparative step to ensure that stage 2 functions well. I argue that *interleaved unlearning* does not solely rely on heuristic thresholds. The confidence threshold used for the poisoned module is one of the many methods that could be used to isolate poisoned data, which I will make clear in the updated manuscript. Interleaved unlearning could be effective without poisoned module. For example, if we know the identities of 80% of backdoored images using Doan et al’s method [3], interleaved unlearning can still be applied using the known backdoored samples without relying on the poisoned module at all.

---

> ### Author Response · Authors · 2024-11-13
> **Response to reviewer JxzC (2)**
>
> - **Reviewer's comment**: The proposed method heavily relies on the prediction of the poisoned module. What if the poisoned module itself is attacked by malicious users? How can the proposed method address this scenario?
>   - **Author's response**: ~~I am not sure what you mean by "the poisoned module itself is attacked by malicious users" since in my threat model, the public (i.e., anyone other than the trusted model trainer) only has the ability to provide the model trainer with potentially backdoored data. The poisoned module is hidden from attackers in my threat model, which is a commonly used threat model and not an outlier compared to existing work (see scenario 2 in Appendix D of [1]). I do not see a specific attack that is applicable to the poisoned module within my threat model. I would be happy to address a specific attack scenario if you elect to provide one.~~  **See my comment below**.
>
> - **Reviewer's comment**: The proposed method appears to be sensitive to the heuristic threshold $c_{thresh}$. How do the authors determine the value of the threshold?
>   - **Author's response**: I would like to point out that in Table 6 of the original manuscript, I show that IEU is (mostly – please read further) not sensitive to the confidence threshold. I did not tune the confidence threshold for CIFAR10 and TinyImageNet (in Table 11 in the appendix, I indicate that I chose 0.95 during my first ever run of IEU and never changed it for TinyImageNet/CIFAR10). No hyperparamter tuning was performed on CIFAR10 and TinyImageNet. Choosing $c_{thresh}$ for GTSRB was more challenging. I used a very rudimentary method for deciding on the confidence threshold: I trained a poisoned module on GTSRB using BadNets-white as the attack and looked at the distribution of the maximum class probability calculated on training images. I manually looked at the deciles of the maximum class probability values and observed that 0.998 led to a low false positive rate (i.e., not too many clean data was being classified as backdoored data). I think looking for a good $c_{thresh}$ value is completely acceptable since I optimise for hyperparameters in the SOTA methods that I compare my method against.
>
> I look forward to your response!
>
> [1] Trap and Replace: Defending Backdoor Attacks by Trapping Them into an Easy-to-Replace Subnetwork (NeurIPS 2022)
>
> [2] Anti-Backdoor Learning: Training Clean Models on Poisoned Data
>
> [3] Defending Backdoor Attacks on Vision Transformer via Patch Processing

---

> ### Author Response · Authors · 2024-11-17
> **Follow-up on Question 1**
>
> - **Reviewer's comment**: The proposed method heavily relies on the prediction of the poisoned module. What if the poisoned module itself is attacked by malicious users? How can the proposed method address this scenario?
>   - **Author's response**: in my initial response, I didn't quite understand what you were referring to with this comment. However, I now think that you are referring to my method's performance when under an adaptive attack where the attacker knows that I am applying the IEU defense. **Am I correct to assume that you're referring to an adaptive attack?**
>
> I will be providing experimental results that show IEU's performance when faced with an adaptive attack similar to the one shown in  "Towards Reliable Backdoor Attacks on Vision Transformers" (https://openreview.net/pdf?id=MLShfiJ3CB).

---

> ### Author Response · Authors · 2024-11-23
> **Looking forward to your response!**
>
> Dear Reviewer JxzC,
>
> I have provided responses and experimental data to address your concerns. You can see my updated manuscript, which you hopefully find is an improvement compared to the initial version.
>
> In light of my updates, would you consider updating your score or updating this thread with further questions?
>
> Thank you!

---

> > ### Comment · Reviewer_JxzC · 2024-11-26
> > **Thank You For the Rebuttal**
> >
> > Thank you for your reply. However, my concerns regarding the quality of writing and novelty have not been well addressed. I would like to keep my original score.

---

### Official Review · Reviewer_5N2c · 2024-11-04

**Soundness:** 3
**Presentation:** 3
**Contribution:** 4
**Rating:** 8
**Confidence:** 4

**Summary:**

This paper introduces Interleaved Ensemble Unlearning (IEU), a defense strategy designed to protect Vision Transformers (ViTs) (or any model in general) from backdoor attacks during the fine-tuning process.

The IEU approach uses a shallow ViT model as a 'poisoned module' to detect backdoored data and protect the main 'robust module,' which is fine-tuned on clean data sorted through the 'poisoned module'.
IEU operates in two stages: the first trains the poisoned module on backdoored data, while the second uses it to isolate potentially poisoned samples for unlearning by the robust module. By alternating between learning and unlearning, the IEU method mitigates the effects of poisoned images, maintaining high accuracy on clean data while effectively defending against backdoor attacks.

Evaluation results on three datasets (CIFAR10, GTSRB, and TinyImageNet) demonstrate its competitive performance against other state-of-the-art backdoor defenses, especially for CIFAR10 and TinyImageNet.

**Strengths:**

- IEU provides a novel approach to mitigating backdoor attacks in ViTs by implementing a layered defense strategy that separates poisoned data using a shallow 'poisoned module'.

- The method’s two-stage process of alternating between learning and unlearning adds robustness against various backdoor attacks without relying on a pre-identified clean set.

- The paper provides comprehensive empirical evaluation against a wide array of backdoor attacks across multiple datasets (CIFAR10, GTSRB, and TinyImageNet), showcasing improvements in ASR reduction and maintenance of high clean accuracy on CIFAR10 and TinyImageNet.

- IEU demonstrates practical value, particularly in domains where ViTs are deployed for security-sensitive tasks, making it a valuable contribution to backdoor defense for ViTs.

- The paper also discusses the method's limitations (e.g., simple dataset, weak attacks, and instability) to provide insights into how to improve the method further in future works.

**Weaknesses:**

- **Limited Explanations for reproducibility:** The article provides limited clarity in the explanations of the technical details, particularly in the specific architectures and configurations used for the 'poisoned module' and 'robust module', respectively. This provides limited insight into understanding the 'poisoned module,' which is the most important component of the proposed approach.

- **Generalizability to Complex Architectures:** It is unclear whether the proposed approach is generalizable to more complex ViT Architectures or is limited to standard ViT architectures. Further, while the approach is proposed specifically for the ViT, in theory, it can be applied to non-ViT-based models, improving the applicability of the proposed defense.

- **Limited insight on the limitations:** The article highlights the limitations and discusses them, providing insights on why the approach underperforms in certain cases. I am not strongly convinced that this is the only reason for underperformance (though it is minimal underperformance). For example, the authors argue that the GTSRB dataset is less complex than the other datasets evaluated, which is why underperformance happens. However, it is a failure case of a 'poisoned module' not to learn well enough to distinguish (as evidenced by Fig 2).
Moreover, the Clean Accuracy is also lower, suggesting that $m_{\theta_p}$, which is directly related to the logits calculated by $f_p$, also affects the performance.
Further, no insights have been provided as to why the poisoned module learn differently for the GTSRB dataset, nor have any suggestions on improving the training of the 'poisoned module' to prevent these failure cases. While I do not state that the approach should perform better for the GTSRB dataset natively, I think there is an oversight in identifying and discussing why the 'poisoned module' gives high enough logits for the poisoned images, which affects both Clean Accuracy as well as ASR directly.

**Questions:**

- Could the authors elaborate on the architectural choices for the shallow ViT used as the poisoned module?

- How does IEU perform on deeper ViT architectures for the 'poisoned module' not covered in the study?

I notice some sharp drops in ASR in Table 2. For Example, for ISSBA-CIFAR-10, the ASR is 100.00 for 0.2 and 0.0 for 0.5. Similar results can be for Smooth and Tiny ImageNet. Can the authors provide some insights on the sharp drops? This could provide better insights for future works to reproduce the results (or even improve).

---

> ### Author Response · Authors · 2024-11-13
> **Response to reviewer 5N2c**
>
> It’s encouraging to see that you liked my paper!
> - **Reviewer's comment**: Limited Explanations for reproducibility: The article provides limited clarity in the explanations of the technical details, particularly in the specific architectures and configurations used for the 'poisoned module' and 'robust module', respectively. This provides limited insight into understanding the 'poisoned module,' which is the most important component of the proposed approach.
>   - **Author's response**: I will provide anonymised code as soon as possible but will explain the architectural details here. The poisoned module is a standard ViT that is very shallow (other architectures can be used but I elected to consider a shallow ViT consisting of ViT block in my paper). The shallow ViT is just like any other ViT: there’s a patch embedding module to produce embeddings for each patch; a classification token is appended to the patch embeddings; then, the patch embeddings of the previous ViT block are passed through subsequent ViT blocks, each of which comprises an attention module and an MLP (one ViT building block by default in my poisoned module); finally, the output of the last ViT block is used by the classification head to produce logits. The robust module is an arbitrary ViT/CNN. The configurations of each are shown in appendix A.2 (table 11). The code for the poisoned/robust modules is standard PyTorch code for the vanilla vision transformer architecture. I will make this clearer in the appendix.
>
> - **Reviewer's comment**: Generalizability to Complex Architectures: It is unclear whether the proposed approach is generalizable to more complex ViT Architectures or is limited to standard ViT architectures. Further, while the approach is proposed specifically for the ViT, in theory, it can be applied to non-ViT-based models, improving the applicability of the proposed defense.
>   - **Author's response**: In practice, IEU can be applied to non-ViT-based models! I provide evidence for generalisability of my IEU defence in Table 8, which shows results of IEU for different defended model architectures, including ViT variants as well as CNNs. I hope this is sufficient to demonstrate the wide applicability if IEU.

---

> ### Author Response · Authors · 2024-11-13
> **Response to reviewer 5N2c (2)**
>
> - **Reviewer's comment**: Limited insight on the limitations: The article highlights the limitations and discusses them, providing insights on why the approach underperforms in certain cases. I am not strongly convinced that this is the only reason for underperformance (though it is minimal underperformance). For example, the authors argue that the GTSRB dataset is less complex than the other datasets evaluated, which is why underperformance happens. However, it is a failure case of a 'poisoned module' not to learn well enough to distinguish (as evidenced by Fig 2). Moreover, the Clean Accuracy is also lower, suggesting that $m_{\theta_p}$, which is directly related to the logits calculated by $f_p$, also affects the performance. Further, no insights have been provided as to why the poisoned module learn differently for the GTSRB dataset, nor have any suggestions on improving the training of the 'poisoned module' to prevent these failure cases. While I do not state that the approach should perform better for the GTSRB dataset natively, I think there is an oversight in identifying and discussing why the 'poisoned module' gives high enough logits for the poisoned images, which affects both Clean Accuracy as well as ASR directly.
>   - **Author's response**: thank you for thoroughly considering my limitations section. Indeed, the failure of the poisoned module at detecting backdoored samples is the reason for low performance (high ASR and low CA). A solution is to build a more discerning ‘poisoned module’, which I did not thoroughly explore in my project since I mainly focused on interleaved unlearning as my novel contribution. The clean accuracy is lower since there is too much overlap between the maximum class probability values of clean and backdoored data regardless of which $c_{thresh}$ I choose (since the unlearn set unfortunately includes clean data in GTSRB, benign representations are unlearnt). Here’s a possible substitution: the backdoor detection method in Doan et al [1] provides a good substitute for my poisoned module and can be integrated with my interleaved unlearning process so that undetected backdoor samples are not learnt. I did not specifically implement Doan’s backdoor detection method in my manuscript since they have fully demonstrated their method’s effectiveness with ViTs. **However**, I made sure to investigate how interleaved unlearning is affected by different true positive rates (positive == backdoor sample, so TPR represents how wel true backdoor samples are being classified as backdoor samples) in table 2. Additionally, I show that my method *is* effective even when backdoored data isn’t detected with high precision in table 6 (see the ‘poison’ and ‘clean’ columns). I admit that I currently (i.e., on the first day of rebuttal) do not have the ability to provide an extremely convincing and more insightful explanation on why the poisoned module fails to distinguish backdoored from clean data in GTSRB. However, I can propose a solution that does not involve replacing the poisoned module with Doan et al’s defence (this is not backed up by data, but I will do the experiments!): the goal of the poisoned module is to learn the backdoored samples very well and *should* be orthogonal to the LGA/gradient flooding methods used in ABL [2], which in essence stops more difficult-to-learn data from being learnt at all (I tried the combination of poisoned module and LGA on CIFAR10 but that did not work, as shown in Table 15 in my appendix C); since the GTSRB dataset is more easily learnt, I *may* need to use local gradient ascent (LGA) or gradient flooding (see [2]) **in conjunction with** my poisoned module to further ensure that clean data in GTSRB is not learnt.
> My hunch was that shortcut learning fails to discern backdoored and benign data for GTSRB because many benign images in GTSRB are basically like backdoors: if the label is “stop sign” then there’s invariably a red circle/octagon/hexagon with a white bar inside of it (or “STOP” inside of the circle). Images in each class has very distinct and obvious attributes in GRSRB, whereas in CIFAR10 and TinyImageNet, a car can take on many different forms (cars are 3D objects and there are much more different car designs than there are variants of stop signs). So, relying solely on shortcut learning isn’t effective to filter out backdoored images for GTSRB.
> I hope this long-winded explanation helps. Please feel free to point out weaknesses in my response so that I can further improve my framework.

---

> ### Author Response · Authors · 2024-11-13
> **Response to reviewer 5N2c (3)**
>
> - **Reviewer's comment**: Could the authors elaborate on the architectural choices for the shallow ViT used as the poisoned module?
>   - **Author's response**: I *duplicated* the first ViT block, the patch embedding module, and classification head of the robust module to form my shallow ViT. Practically, this is a convenient and effective choice: ViTs are more modularised than CNNs, meaning I can directly feed the output of the first ViT block into the classification head without making modifications (I also had a readily available ViT checkpoint, so it’s convenient in terms of writing code to use part of it as the shallow poisoned module). In addition, it’s an intuitive choice: a shallow module is very likely to learn shortcuts in the dataset very well (e.g., backdoored image and its label form a shortcut) without getting too confident when facing benign features.
> - **Reviewer's comment**: How does IEU perform on deeper ViT architectures for the 'poisoned module' not covered in the study?
>   - **Author's response**: I change the depth of the shallow poisoned ViT as part of my ablation study, but I stop at depth == 3 since CA already becomes unacceptably low at depth == 3. I suspect that IEU will perform worse (low CA but ASR does not change) as the depth of the poisoned module increases since a deeper poisoned module also encourages the learning of non-shortcut samples. This causes the unlearn set to have an excessive amount of clean data which is not the desired outcome.
> - **Reviewer's comment**: I notice some sharp drops in ASR in Table 2. For Example, for ISSBA-CIFAR-10, the ASR is 100.00 for 0.2 and 0.0 for 0.5. Similar results can be for Smooth and TinyImageNet. Can the authors provide some insights on the sharp drops? This could provide better insights for future works to reproduce the results (or even improve).
>   - **Author's response**: I suspect at $\alpha_i \div \alpha=0.2$ (replace \alpha_i with \hat{\alpha_i} since \hat{\alpha} isn’t working), there isn’t enough poisoned data inside of the unlearn set $D^{ul}$ for the unlearning to succeed in erasing the more hidden and difficult-to-remove attacks (ISSBA and Smooth). The association between ISSBA’s backdoored samples and the label are harder to remove due to its stealthiness and lack of a sharp spike in attention in one specific location of the image space (ISSBA and Smooth are both patterns that cause the values in the attention interpretation map to be distributed across the entire image space, whereas images with the BadNets-white backdoor exhibit high attention around the trigger when using GradRollout [4] to interpret the attention). For BadNets-white, this effect is not as significant since there is a clearer association between the more obvious backdoor trigger and the label, meaning that unlearning the more obvious triggers of BadNets-white only requires a few samples. In other words, the model sees an obvious feature to unlearn when unlearning images with BadNets-white triggers. At $\alpha_i \div \alpha=0.5$, more pernicious and stealthier attacks can be erased more effectively since the robust module unlearns more backdoored samples. It's difficult to efficiently pinpoint the neurons that activate thanks to more hidden attacks like ISSBA and Smooth (see neuron cleansing defences). In addition, ISSBA & Smooth lead to weaker shortcuts in the robust module. Weaker shortcuts are more difficult to learn due to the weaker association between a feature in the backdoored image and the target label. Once the weak association between the hidden backdoor trigger and the target label is implanted in the model, the non-obvious connection between the backdoor trigger and the target label has a similarly weak effect during unlearning. Hence, more backdoor data is needed to unlearn ISSBA and Smooth.
>
> [1] Defending Backdoor Attacks on Vision Transformer via Patch Processing
>
> [2] Anti-Backdoor Learning: Training Clean Models on Poisoned Data
>
> [3] Trap and Replace: Defending Backdoor Attacks by Trapping Them into an Easy-to-Replace Subnetwork
>
> [4] https://jacobgil.github.io/deeplearning/vision-transformer-explainability

---

> ### Author Response · Authors · 2024-11-17
> **Insight on limitations**
>
> |           | ('CIFAR10', 'BadNets-white')   | ('CIFAR10', 'ISSBA')   | ('TinyImageNet', 'BadNets-white')   | ('TinyImageNet', 'ISSBA')   | ('GTSRB', 'BadNets-white')   | ('GTSRB', 'ISSBA')   |
> |:----------|:-------------------------------|:-----------------------|:------------------------------------|:----------------------------|:-----------------------------|:---------------------|
> | LGA       | 0.00/99.98                     | 0.00/65.84             | 0.03/20.07                          | 0.12/67.32                  | 11.51/22.07                  | 8.38/34.89           |
> | no-method | 5.62/5.30                      | 5.58/2.10              | 0.21/11.97                          | 0.15/22.59                  | 44.72/14.86                  | 42.63/5.71           |
>
> The above table compares the ability of the poisoned module to distinguish between clean and backdoored data for all three datasets. The values shown in each cell are fnr/fpr (percentages) where backdoored images belong to the positive class. This table shows that LGA is not very effective when applied on CIFAR10/TinyImageNet, but can be very effective on GTSRB as shown by the large decrease in fnr when using LGA compared to when not (decreases from 44.72 to 11.51).
>
> I don't think I adequately addressed your comments on the limitations. With more experiments that demonstrate IEU's performance on GTSRB when tuning stage 1 with LGA/Flooding, I can provide a more in-depth treatment.
>
> As explained in [1], LGA (local gradient ascent) maximises the gradient around a fixed loss value $\gamma$. This means that, if the loss of a particular mini-batch dips below $\gamma$, then gradient ascent is performed on that mini-batch (in ABL's paper they seem to indicate that this is done per-image in the "Backdoor Isolation" paragraph of section 3.2 in https://arxiv.org/pdf/2110.11571, but the code indicates that LGA is done per mini-batch). Since the loss values of backdoor samples drop significantly faster, they will escape the gradient ascent which is at its highest strength around $\gamma$. Recall that LGA performs gradient descent on $lgaLoss$ where it’s defined as $lgaLoss = sgn(loss - \gamma) \cdot loss$ (where $loss$ is the normal cross entropy loss). This means that when $loss<\gamma$, gradient *ascent* is performed. Although ABL does not explain why, I think I can give a good explanation for why backdoor samples don’t get unlearned with LGA when their loss is below $\gamma$: if we assume that backdoor samples do indeed drop below the threshold $\gamma$ much more quickly, then loss on backdoor samples will be either above $\gamma$ or significantly below $\gamma$ (i.e., close to zero); local gradient ascent performs gradient ascent for *any* $loss < \gamma$, but results in a less significant update with backdoored samples with very low loss (i.e., the model’s parameters are close to the optimal values for these backdoored samples) since the gradient is very small at a local minimum. This is similar to how gradient descent results in diminishing decrease in loss value as the model parameters approaches a local minimum. This is why backdoor samples will be trapped.
>
> Back to the original issue of why poisoned module for GTSRB didn’t work well without LGA/Flooding. I think the more easily learned GTSRB dataset can adequately explain the increase in performance with LGA. In CIFAR10 and TinyImageNet, the clean images did not get the chance to be learned properly by the poisoned module. With LGA, the poisoned module did not learn much benign features when applied on GTSRB. This *suggests* that without LGA, the poisoned module probably *did* learn a lot of benign features on GTSRB and did not focus on solely learning backdoored features. This divides the focus of the module between learning benign and backdoored samples and I think explains why the backdoored images (ISSBA attack) were not being learned properly for GTSRB. Comparing GTSRB with CIFAR10/TinyImagenet (where LGA was not required), I think a plausible explanation is that GTSRB’s relative simplicity caused the poisoned module to learn more benign features. LGA separates low-loss and high-loss samples in a backdoored dataset where the distinction isn’t so clear (such as GTSRB), which is why it worked with GTSRB.
>
> **Please do feel free to ask more questions.**
>
> [1] Anti-Backdoor Learning: Training Clean Models on Poisoned Data

---

> ### Author Response · Authors · 2024-11-23
> **Looking forward to your response!**
>
> Dear Reviewer 5N2c,
>
> Thank you for your review! I have addressed all your concerns in my responses in this thread. I hope you have viewed the additional experiments that I have provided and are satisfied with the new version of the manuscript. Please update the thread with any lingering comments if you have any.
>
> Thanks!

---

### Official Review · Reviewer_JLmi · 2024-11-09

**Soundness:** 3
**Presentation:** 2
**Contribution:** 2
**Rating:** 5
**Confidence:** 4

**Summary:**

This paper presented Interleaved Ensemble Unlearning (IEU), a method for finetuning
clean ViTs on backdoored datasets. IEU includes two stages, where the first one is designed to train a shallow ViT used to block potentially poisoned data and the second stage defends backdoor attacks utilizing unlearning. The experiments demonstrate that IEU out-performs existing defenses on diverse datasets and backdoor attacks.

**Strengths:**

-	The two-stage method is reasonable, especially, using a shallow model to learn shortcuts in the dataset in the first stage.
-	The experiments show that IEU performs better than existing methods, including I-BAU and ABL. Besides, the ablation studies are well-organized, illustrating the necessity of designs in IEU.

**Weaknesses:**

-	The novelty is limited. The proposed IEU utilizes the unlearning strategy for backdoor defense. Compared to ABL, the main differences lie in using a shallow model to block potentially poisoned data and a confidence threshold to determine the unlearned samples rather than a fixed-sized unlearned set.

-	I am confused as to why IEU is tailored to ViTs. According to my understanding, there is no customized design in IEU for transformer-like architectures. Also, the authors evaluate CNN models in Table 8. Hence, if I am correct, I recommend the authors revise the writing to highlight the universality of the proposed IEU.

-	I suggest that Eq.2 could be further clarified in detail. If the logits for optimizing the objective in Equation 2 sometimes come from $f_p$, should the optimized parameters include $\theta_p$?

**Questions:**

Please see the weakness part.

---

> ### Author Response · Authors · 2024-11-13
> **Response to reviewer JLmi**
>
> - **Reviewer's comment**: The novelty is limited. The proposed IEU utilizes the unlearning strategy for backdoor defense. Compared to ABL, the main differences lie in using a shallow model to block potentially poisoned data and a confidence threshold to determine the unlearned samples rather than a fixed-sized unlearned set.
>   - **Author's response**: I would like to emphasise that the novelty of my backdoor defense (which I may have not stressed enough in the paper) comes from the **interleaved unlearning** where the shallow model is only the secondary contribution (I gave credit to DenoisedPoE [1] for inspiring this defence). The interleaved unlearning process encourages the backdoor features to be forgotten during unlearning, but the unlearning process is then immediately followed by learning on mostly clean data to prevent benign features from being forgotten. This is one of the major innovations and improvements compared to previous unlearning-based methods: instead of unlearning some backdoored data after training/tuning which causes the model to forget benign features, alternating between unlearning backdoored data and learning clean data means that the clean data is not likely to be forgotten. The alternating optimisations presented in equations 2 and 4 of section 3.1 will be combined into one equation for clarity. In addition, I will make sure to emphasise the interleaved part of my framework as the main contribution.
> - **Reviewer's comment**: I am confused as to why IEU is tailored to ViTs. According to my understanding, there is no customized design in IEU for transformer-like architectures. Also, the authors evaluate CNN models in Table 8. Hence, if I am correct, I recommend the authors revise the writing to highlight the universality of the proposed IEU.
>   - **Author's response**: I agree that IEU does not have customised designs for transformer-like architectures; however, the performance of IEU is much better compared to other non-ViT-specific methods when defending the ViT. When I designed this framework, specifically the poisoned module which I more-or-less duplicate from the defended ViT, I thought that CNNs are less amenable to my IEU since CNNs are less easily modularised: the dimensions of the intermediate convolutional layers' outputs are not always compatible with the original classification head's linear layers. For example, the seminal ResNet's intermediate features increase in dimensions as we go from the first layer to the last layer. However, for *most* ViT variants, each attention+MLP layer's output is the same shape regardless of whether we consider the first or last layer's outputs. This is my reasoning for suggesting that my IEU is more compatible for ViTs (and is very briefly mentioned on line 053). Combined with the superior performance of my IEU when used for ViTs, I argue that IEU is designed for ViTs despite the fact that IEU does not address a specific vulnerability in ViTs. I will highlight both the superior performance of IEU when defending ViTs as well as its potential for universality when applied to different architectures. Thank you for this valuable comment.
> - **Reviewer's comment**: I suggest that Eq.2 could be further clarified in detail. If the logits for optimizing the objective in Equation 2 sometimes come from $f_p$, should the optimized parameters include $\theta_p$?
>   - **Author's response**: Thank you for the suggestion. In equation 1, $y = y_p ( 1-m_{\theta_p} ) + y_r m_{\theta_p}$ (not sure how to do \hat{y}, but you can substitute y with \hat{y} when reading this expression), which means that the logits used for loss and back-propagation is calculated using a linear combination of the logits from the poisoned (p) and robust (r) module, respectively. This indeed means that the logits for optimising the objective in Equation 2 do occasionally come from $f_p$. However, during stage 2, only the robust module is being optimised as indicated in Figure 1 by the “lock” icon. This means that I am not optimising the parameters of $f_p$ in stage 2. I decided to freeze $\theta_p$ in stage 2 since its main function is to have high confidence on backdoored data, which is already achieved during stage 1. Hence, there is no need to update $\theta_p$ during stage 2.
>
> I look forward to your response!
>
> [1] From Shortcuts to Triggers: Backdoor Defense with Denoised PoE
>
> [2] Anti-Backdoor Learning: Training Clean Models on Poisoned Data

---

> ### Author Response · Authors · 2024-11-23
> **Looking forward to your response!**
>
> Dear Reviewer JLmi,
>
> I have addressed all your concerns via responses to this thread, updates to the manuscript, and additional experimental data. If you have any lingering questions, please post them in this thread! I hope that you will reconsider your rating in light of my updates during the rebuttal.
>
> Thank you!

---

> ### Author Response · Authors · 2024-11-30
> **Would you like to respond to my rebuttal?**
>
> Hi!
>
> I hope you'll allocate some time to reading and responding to my rebuttal.
>
> Here's a summary of improvements and new experiments:
> - The updated manuscript is hopefully easier to read.
>   - Figure 1 has been edited to include better explanations for expressions
>   - The related work section has been updated to be more thorough with a new paragraph on unlearning. Additionally, the related work section includes two more references and introduces backdoor attacks/defences with more clarity.
> - I have produced experiments using a new setting (specifically, tuning the poisoned module using Local Gradient Ascent) that shows that IEU can defend backdoor attacks using the GTSRB dataset. The GTSRB results in the main results table (Table 3) have been updated accordingly.
> - I have implemented an adaptive attack and demonstrated that IEU is able to fully defend against this adaptive attack (see Appendix D and my post titled "The new attack does exactly what you've requested! So, it's an adaptive attack").
> - I have re-organised the limitation and discussion section as well as the appendix to better reflect the updated experimental results for GTSRB.
>
> To address your concerns, I have emphasised the novelty of the IEU approach in the manuscript (sections 1 and 3) as well as in previous comments posted directly to this thread. I've also re-phrased the contributions of IEU in a clearer way that does not inaccurately claim that IEU is designed specifically for ViTs (see section 1 in the manuscript) and I've also emphasised the universality of IEU. I addressed your concern about Eq 2 in a previous comment (tl;dr: only the robust module is being optimised in stage 2, so $\theta_p$ is not included as an optimisation parameter).
>
> Thank you for your feedback and I hope you will reconsider your rating!

---

> ### Author Response · Authors · 2024-12-03
> **Just a few hours left until the discussion period ends!**
>
> Hi (and sorry for pinging you yet again), there's a few hours left before the discussion period ends. If you make a post with questions/things that need to be clarified before the discussion period ends, I will have one day (until 23:59 Dec 3rd AOE) to address any questions/requests. I'd love to read your feedback regarding the rebuttal!
>
> Thank you for your review!

---

### Author Response · Authors · 2024-11-13
**Initial comments and plans**

Based on the reviewers’ insightful suggestions, I aim to make the following modifications to my manuscript:
- I will edit Figure 1 and its caption so that symbols are properly explained.
- I will emphasise my main contributions, which should pinpoint the **interleaved unlearning** framework as the main, novel contribution. The novel process of interleaved unlearning encourages the model to forget backdoor features without leading to a large drop in clean accuracy. This alternating updates (slightly similar to ADMM) towards optimising for two different objectives (unlearning backdoors and learning clean data) is a novel and significant improvement over SOTA methods.
- I plan to include more well-written background information on backdoor attacks. Currently, the clarity of the first part of my introduction can especially be improved. I will also include a few more papers in my literature review.
- I will de-emphasise my rather flimsy argument that IEU is *specifically* designed for ViTs and will instead stress that its performance on ViTs is better than existing SOTA methods. Additionally, the main contribution is the interleaved unlearning framework and not the distracting and slightly spurious argument that IEU is specifically for ViTs.
- I will provide more insights on the limitations of my model.
- I will emphasise that the poisoned module is one way of detecting backdoor samples and can be replaced by other methods for backdoor image detection because the novel contribution of my method should *not* be misconstrued as being the poisoned module (it is instead the **interleaved unlearning** framework). Since backdoor image detection methods are well-developed and can isolate backdoored images with high precision, I simulated situations where certain percentages of backdoored samples are detected (see table 2 in section 3.2).
- I will improve the flow of my paper in general and repurpose existing tables to emphasise different messages.

Additionally, I will produce additional results:
- I aim to design an adaptive attack for IEU.
- I will produce results for different target labels (e.g., results where the backdoored class $\neq 1$).
- I will investigate how (and whether) LGA/gradient flooding (see ABL [1]) can improve the performance of the poisoned module to address the limitations I presented in section 5 of my paper.

A clean version of the IEU code will be provided as a link to an anonymous repository via a non-public post.

I sincerely thank reviewers for dedicating their time to reading and reviewing my project and I am very, very grateful to receive the first round of feedback. I look forward to discussing my paper with all of you and will endeavour to improve my paper!

[1] Anti-Backdoor Learning: Training Clean Models on Poisoned Data

---

### Author Response · Authors · 2024-11-16
**Ablation study: different trigger labels (thanks to reviewer NEex's suggestion)**

|  trigger label  |   ('BadNets-white', 'ASR') |   ('BadNets-white', 'CA') |   ('ISSBA', 'ASR') |   ('ISSBA', 'CA') |   ('Smooth', 'ASR') |   ('Smooth', 'CA') |
|---:|---------------------------:|--------------------------:|-------------------:|------------------:|--------------------:|-------------------:|
|  0 |                       0.71 |                     92.28 |               0.1  |             98.43 |                0.07 |              97.71 |
|  1 |                       0.96 |                     98.19 |               0.33 |             98.35 |                0.09 |              97.77 |
|  3 |                       0.51 |                     97.88 |               0.03 |             98.36 |                0.1  |              93.87 |
|  5 |                       0.37 |                     97.53 |               0    |             98.35 |                0.05 |              97.71 |
|  8 |                       0.19 |                     94.51 |               0    |             98.14 |                0.01 |              97.51 |


The table above shows the ASR and CA (percentage) when using different trigger labels on the CIFAR10 dataset for three different attacks. Note that only a subset of trigger labels are considered due to compute limitations. The trigger label used in the main body of the paper is trigger label == 1. This addresses reviewer NEex's recommendation to conduct "further experiments to assess whether BSD (Backdoor Sample Detection) can successfully defend against backdoor attacks with different target labels".

The CA is mostly high and the ASR is consistently low for all attacks. This suggests that my IEU is able to defend against attacks with different target labels.

This table will appear in the appendix of the revised manuscript.

---

### Author Response · Authors · 2024-11-16
**IEU successfully defends ViTs on the GTSRB dataset**

|          |   ('BadNets-white', 'ASR') |   ('BadNets-white', 'CA') |   ('ISSBA', 'ASR') |   ('ISSBA', 'CA') |   ('Smooth', 'ASR') |   ('Smooth', 'CA') |
|:---------|---------------------------:|--------------------------:|-------------------:|------------------:|--------------------:|-------------------:|
| Flooding |                   2.20111  |                   85.3761 |            2.69992 |           86.9913 |            20.3642  |            87.8622 |
| LGA      |                   2.21694  |                   83.2621 |            2.81077 |           86.4766 |            15.3682  |            88.4481 |
| Neither  |                   0.926366 |                   82.0744 |          100       |           93.5234 |             6.15202 |            84.521  |

The above table shows the performance of IEU on GTSRB when using Loss Flooding, local gradient ascent (LGA), or neither ([1]) when tuning the poisoned module in stage 1. The Flooding/LGA threshold parameter $\gamma$ is set to 1. Note that $c_{thresh} = 0.9$ is used instead of $c_{thresh} = 0.95$. The stage 1 settings for GTSRB follow CIFAR10/TinyImageNet (note that GTSRB settings in this table and the GTSRB settings I'm planning to use in the revised manuscript are different from the GTSRB settings I used in the pre-rebuttal's manuscript: for GTSRB pre-rebuttal, I used 1e-3 as the prefinetune_lr and only tuned the poisoned module for 5 epochs; here, I'm using 2e-4 as the prefinetune_lr and tuned the poisoned module for 10 epochs, meaning that all settings *except* for $c_{thresh}$ for GTSRB are identical to CIFAR10/TinyImageNet). Loss Flooding and LGA are designed to induce the poisoned module to learn shortcuts.

Inspired by reviewer 5N2c's comments about IEU's performance on GTSRB, I wanted to see whether LGA/Flooding would improve the model's performance on GTSRB. It turns out that both LGA and Flooding allow the poisoned module to be successfully backdoored without learning much benign features when used separately (meaning that I do not apply both LGA and Flooding in the same experiment) to tune stage 1's poisoned module for GTSRB.

Since I can use Flooding/LGA to tune a defended ViT in stage 2 when using GTSRB, (most) **weaknesses in the limitations section (sec 5 in the pre-rebuttal paper) have been addressed** and I will update the text and tables in my revised manuscript accordingly.

[1] Anti-Backdoor Learning: Training Clean Models on Poisoned Data

---

### Author Response · Authors · 2024-11-20
**IEU successfully defends against an adaptive attack**

**In response to the comments from reviewers JxzC and NEex**, I have implemented an adaptive attack designed to induce the poisoned module to mis-predict.


|                     |   ('IEU', 'ASR') |   ('IEU', 'CA') |   ('No Defense', 'ASR') |   ('No Defense', 'CA') |
|:--------------------|-----------------:|----------------:|------------------------:|-----------------------:|
| CAT+BadNets-pattern |             0    |           98.27 |                  100    |                  97.77 |
| CAT+BadNets-white   |             0.74 |           94.9  |                   94.72 |                  98.31 |
| CAT+Blended         |             0    |           98.28 |                   99.96 |                  98.38 |
| CAT+Smooth          |             0.1  |           97.87 |                   98.88 |                  98.33 |
| CAT+Trojan-SQ       |             0.02 |           98.25 |                   99.66 |                  98.44 |

This [1] adaptive attack called "CAT" is used and the table above shows the results of this attack when the model trainer uses IEU (left two columns) and when the model trainer uses no defence (right two columns) on CIFAR10. IEU **demonstrates high performance (low ASR and high CA) on all attacks+CAT** except for Blended+CAT, where the CA using IEU is $\sim 8$ percentage points lower than the CA without a defence. All settings used for these experiments are the same as the settings I used in the paper (CIFAR10 parameters).

The gist is that CAT optimises the backdoored images so that they create channel activations that are more similar to those found in normal benign images, meaning that CAT is applied on existing backdoor attacks (e.g., BadNets-white) to find stealthier triggers. CAT is evaluated specifically on Vision Transformers in [1] and demonstrates good performance.

This is an adaptive attack for my IEU defense because the CAT perturbations aim to make the poisoned module misclassify the poisoned images by encouraging channel activations for benign and backdoored images to be similar, which aims to cause the poisoned module to misclassify images.

Since the authors of [1] have not released their code, I implemented their attack from scratch under a gray-box scenario, where the attacker has access to a tuned version of the poisoned module. First, the attacker trains two classifiers (the backdoor discriminator and the target classifier) using the features generated by the poisoned module (specifically, the output of the MLP layer before the classification head in my shallow poisoned module). Then, these two classifiers (I used shallow ViTs with depth=4 as the two classifiers) along with the poisoned module are used to generate backdoored data with adversarial perturbations applied onto the backdoor data via projected gradient descent (e.g., some $\delta'$ is added onto the backdoored image $x+\delta$). Then, the backdoored images with CAT adversarial perturbations are used to train a defended ViT using IEU from stage 1. Since stage 1 and stage 2 are bundled together, I train a new poisoned module on the backdoored data generated from CAT.

[1] Towards Reliable Backdoor Attacks on Vision Transformers

---

> ### Author Response · Authors · 2024-11-21
> **Update: adaptive attack results for TinyImageNet**
>
> |                     |   ('IEU', 'ASR') |   ('IEU', 'CA') |   ('No Defense', 'ASR') |   ('No Defense', 'CA') |
> |:--------------------|-----------------:|----------------:|------------------------:|-----------------------:|
> | CAT+BadNets-pattern |             0    |           68.32 |                  100    |                  69.77 |
> | CAT+BadNets-white   |             0.04 |           69.43 |                   95.96 |                  70.09 |
> | CAT+Blended         |             0    |           65.58 |                   99.97 |                  70.46 |
> | CAT+Smooth          |             0.01 |           68.1  |                   92.47 |                  70.34 |
> | CAT+Trojan-SQ       |             0    |           66.66 |                   99.63 |                  70.71 |
>
> Same table as above, but for TinyImageNet.
>
> Conclusion: IEU defends against grey-box attacks.
>
> - Why is this a grey-box attack in this context?
>   - The attacker has access to a version of $f_p$ that the defender has trained using the same dataset, the same hyperparameters, the same data augmentations, and the same training setup as the defender uses during normal defender-specified applications of IEU; however, the attacker is not in control of stage 1 or stage 2. This is consistent with the threat model that I use (attacker can provide malicious data but cannot control the training/inference process).

---

### Author Response · Authors · 2024-11-23
**Uploaded new version of the paper**

Dear Reviewers and AC,

Thank you all for the reviews. They've been very helpful for me when trying to improve the paper's writing and completeness.

I've uploaded a new version of the paper which, in tandem with my other responses, should address all concerns by reviewers.

Specifically, I address the following concerns:
- Reviewers **JLmi** and **JxzC**: concern about the novelty of IEU.
  - The updated manuscript shifts the focus to the novel Interleaved Unlearning framework as my main, novel contribution.
- Reviewer **JxzC**: concern about presentation and clarity.
  - The related work section how includes a paragraph about unlearning. In addition, the start of the Backdoor Attacks paragraph in the related work section has been edited to include a better introduction for backdoor attacks.
  - Figure 1 has been updated to include explanations of various expressions.
- To address the reviewers’ concerns regarding whether the IEU is specifically designed for ViTs, I have modified the manuscript to replace this claim with a claim that indicates “IEU demonstrates superior performance on ViTs”.
- Reviewer **NEex**: more experiments related to performance on different target labels
  - The relevant experiment has been added to Appendix C (Table 17).
- Reviewr **NEex** and **JxzC**: an adaptive attack (what happens when the poisoned module is attacked?)
  - Appendix D demonstrates that my IEU is robust to an adaptive attack.
- Reviewer **NEex**: useful papers to cite.
  - I have included the two papers in the related works that reviewer NEex suggested.

Additionally, the revised manuscript has been updated to include newest experimental data (specifically, tuning the poisoned module with LGA for GTSRB in table 3, table 15, and table 16). The Discussions and Limitations section has also been revised to reflect the performance of IEU on GTSRB when LGA is used. I have edited the methods section to be more readable.

I look forward to engaging with reviewers through interactive discussions!

---

### Author Response · Authors · 2024-11-23
**Strengths of this submission**

As we approach the end of the discussion period, I would like to emphasise and summarise this submission's strengths.

- All reviewers agree that **strong experimental results** and the ablation study have demonstrated that IEU performs well when defending ViTs.
  - Reviewer **JLmi** praises the ablation study for being “well-organised” and asserts that “the experiments show that [my method] IEU performs better than existing methods”.
  - Reviewer **5N2c** highlights the experimental section for its “comprehensive empirical evaluation”.
  - Reviewer **NEex** provides positive comments regarding my ablation study, writing that it is “organised well to clearly demonstrate the whole proposed method”. Additionally, reviewer NEex indicates that the “good performance obtained by the experiments” further supports the effectiveness of IEU.
  - Reviewer **JxzC** indicates that the paper provides “extensive empirical data on performance improvements”.
- Reviewer **5N2c** notes the **novel** approach that IEU takes and highlighted the fact that IEU does not rely on a clean holdout dataset. My belief is that the Interleaved Unlearning framework, which is the main contribution of this submission, is a novel approach to backdoor defence and the updated manuscript reflects this.
- Reviewers **NEex** and **JxzC** point out that this submission **fills a gap in existing literature**. Reviewer **NEex** indicates that “tailored solutions are limited” (indicating that ViTs do need tailored backdoor defences). Furthermore, reviewer **JxzC** writes that my IEU “address[es] the scarcity of ViT-targeted backdoor defences”.

---

### Author Response · Authors · 2024-11-30
**Reminder to respond to my rebuttal :)**

Hi all and apologies for pinging you again,

While I can't say this rebuttal was built on blood, sweat, and tears, it definitely took a lot of effort and time to design new experiments, implement the adaptive attack, and refine previous results.

It would be great if reviewers could at least confirm that they've had a chance to review my rebuttal.

Since we still have a few days left in the discussion period, additional comments, feedback, or score changes would all mean a great deal to me. I hope my experiments and previous comments have brought more clarity to the IEU framework!

Thanks!

---

### Note · Authors · 2025-01-22

**Comment:**

Thank you for the valuable suggestions!

**Withdrawal Confirmation:**

I have read and agree with the venue's withdrawal policy on behalf of myself and my co-authors.